# Information-Theoretic Analysis of Unsupervised Domain Adaptation

**Ziqiao Wang & Yongyi Mao**
University of Ottawa
{zwang286,ymao}@uottawa.ca

## Abstract

This paper uses information-theoretic tools to analyze the generalization error in unsupervised domain adaptation (UDA). We present novel upper bounds for two notions of generalization errors. The first notion measures the gap between the population risk in the target domain and that in the source domain, and the second measures the gap between the population risk in the target domain and the empirical risk in the source domain. While our bounds for the first kind of error are in line with the traditional analysis and give similar insights, our bounds on the second kind of error are algorithm-dependent, which also provide insights into algorithm designs. Specifically, we present two simple techniques for improving generalization in UDA and validate them experimentally.

## 1 Introduction

This paper focuses on the *unsupervised domain adaptation (UDA)* task, where the learner is confronted with a source domain and a target domain and the algorithm is allowed to access to a labeled training sample from the source domain and an unlabeled training sample from the target domain. The goal is to find a predictor that performs well on the target domain.

A main obstacle in such a task is the discrepancy between the two domains. Some recent works (Ben-David et al., 2006; 2010; Mansour et al., 2009; Zhao et al., 2019; Zhang et al., 2019; Shen et al., 2018; Germain et al., 2020; Acuna et al., 2021; Nguyen et al., 2022) have proposed various measures to quantify such discrepancy, either for the UDA setting or for the more general domain generalization tasks, and many learning algorithms are proposed. For example, Nguyen et al. (2022) uses a (reverse) KL divergence to measure the misalignment of the two domain distributions, and motivated by their generalization bound, they design an algorithm that penalizes the KL divergence between the marginal distributions of two domains in the representation space. Despite that this "KL guided domain adaptation" algorithm is demonstrated to outperform many existing marginal alignment algorithms (Ganin et al., 2016; Sun & Saenko, 2016; Shen et al., 2018; Li et al., 2018), it is not clear whether KL-based alignment of marginal distributions is adequate for UDA, and more fundamentally, what role the unlabelled target-domain sample should play in cross-domain generalization. Notably, most UDA algorithms are heuristically designed and intuitively justified. Moreover, most existing generalization bounds are algorithm-independent. Then there appears significant room for both deeper theoretical understanding and more principled algorithm design.

In this paper, we analyze the generalization ability of hypotheses and learning algorithms for UDA tasks using an information-theoretic framework developed in (Russo & Zou, 2016; Xu & Raginsky, 2017). The foundation of our technique is the Donsker-Varadhan representation of KL divergence (see Lemma A.1). We present novel upper bounds for two notions of generalization errors. The first notion ("population-to-population (PP) generalization error") measures the gap between the population risk in the target domain and that in the source domain *for a hypothesis*, and the second ("expected empirical-to-population (EP) generalization error") measures the gap between the population risk in the target domain and the empirical risk in the source domain *for a learning algorithm*. We show that the PP generalization error for all hypotheses are uniformly bounded by a quantity governed by the KL divergence between the two domain distributions, which, under bounded losses, recovers the the bound in Nguyen et al. (2022). We then show that this KL term upper-bounds some other measures including Total-Variation distance (Ben-David et al., 2006), Wasserstein dis-

tance (Shen et al., 2018) and domain disagreement (Germain et al., 2020). Thus, minimizing KL-divergence forces the minimization of other discrepancy measures as well. This, together with the ease of minimizing KL (Nguyen et al., 2022), explains the effectiveness of the KL-guided alignment approach. For expected EP generalization error, we develop several algorithm-dependent generalization bounds. These algorithm-dependent bounds further inspire the design of two new and yet simple strategies that can further boost the performance of the KL guided marginal alignment algorithms. Experiments are performed to verify the effectiveness of these strategies.

## 2 RELATED WORK

**Domain Adaptation** Many domain adaptation generalization bounds have been developed (Ben-David et al., 2006; 2010; David et al., 2010; Mansour et al., 2009; Shen et al., 2018; Zhang et al., 2019; Germain et al., 2020; Acuna et al., 2021), and various discrepancy measures are introduced to derive these bounds including total variation (Ben-David et al., 2006; 2010; David et al., 2010; Mansour et al., 2009), Wasserstein distance (Shen et al., 2018), domain disagreement (Germain et al., 2020) and so on. In particular, bounds based on $\mathcal{H}\Delta\mathcal{H}$ in Ben-David et al. (2010) are restricted to a binary classification setting and assume a deterministic labeling function. Furthermore, Ben-David et al. (2010) also assumes the loss is the $L_1$ distance between the predicted label and true label (which is bounded). Our bounds work for the general supervised learning problems with any labelling mechanism (e.g., stochastic labelling), and we do not require the specific choice of the loss (even unbounded). Recently, Shui et al. (2020) proposed generalization bounds using Jensen-Shannon (JS) divergence, which bear a relation to our Corollary 4.2. While other algorithm-dependent bounds have been proposed for different transfer learning settings (e.g., Wang et al. (2019)), they are not directly comparable to our own bounds. For more details about the domain adaptation theory, we refer readers to Redko et al. (2020) for a comprehensive survey. In addition, the most common methods for domain adaptation involve aligning the marginal distributions of the representations between the source and target domains, for example, using an adversarial training mechanism (Ganin et al., 2016; Shen et al., 2018; Acuna et al., 2021) or aligning the first two moments of the representation distribution (Sun & Saenko, 2016). There are numerous other domain adaptation algorithms, and we refer readers to (Wilson & Cook, 2020; Zhou et al., 2021; Wang et al., 2021b) for recent advances.

**Information-Theoretic Generalization Bounds** Information-theoretic analysis is usually used to bound the expected generalization error of supervised learning, where the training and testing data come from the same distribution (Russo & Zou, 2016; 2019; Xu & Raginsky, 2017; Bu et al., 2019; Negrea et al., 2019; Steinke & Zakynthinou, 2020; Rodríguez Gálvez et al., 2021). Exploiting the chain rule of mutual information, these bounds are successfully applied to characterize the generalization ability of stochastic gradient based optimization algorithms (Pensia et al., 2018; Negrea et al., 2019; Haghifam et al., 2020; Wang et al., 2021a; Neu et al., 2021; Wang & Mao, 2022a;b). Recently, this framework has also been used in other learning settings including meta-learning (Jose & Simeone, 2021a; Jose et al., 2021; Rezazadeh et al., 2021; Chen et al., 2021), semi-supervised learning (He et al., 2021; Aminian et al., 2022) and transfer learning (Wu et al., 2020; Jose & Simeone, 2021a;b; Masiha et al., 2021; Bu et al., 2022). In particular, (Wu et al., 2020; Jose & Simeone, 2021b) consider a different problem setup with ours. Specifically, their expected generalization error is the gap between the target population risk and a weighted empirical risk combining both the source and the target empirical risks, while our "EP" error is the gap between the target population risk and the source empirical risk. That is, we focus on the role of the *unlabelled* target data in cross-domain generalization when the source empirical risk is taken as a training objective, whereas their works assume the existence of *labelled* target data and study their role in domain adaptation.

## 3 PRELIMINARY

Unless otherwise noted, a random variable will be denoted by a capitalized letter, and its realization is denoted by the corresponding lower-case letter. Consider a prediction task with instance space $\mathcal{Z} = \mathcal{X} \times \mathcal{Y}$, where $\mathcal{X}$ and $\mathcal{Y}$ are the input space and the label (or output) space, respectively. Let $\mathcal{F}$ be the hypothesis space of interest, in which each $f \in \mathcal{F}$ is a function or predictor mapping $\mathcal{X}$ to $\mathcal{Y}$. We assume that each hypothesis $f \in \mathcal{F}$ is parameterized by some weight parameter $w$ in some space $\mathcal{W}$ and may write $f$ as $f_w$ as needed.

Let $\mu$ and $\mu'$ be two distributions on $\mathcal{Z}$, unknown to the learner, where $\mu$ characterizes the source domain and $\mu'$ characterizes the target domain. We may also write $\mu$ as $P_Z$ or $P_{XY}$ and $\mu'$ as $P_{Z'}$ or $P_{X'Y'}$, which defines random variables $Z = (X, Y)$ and $Z' = (X', Y')$, respectively. Let $S = \{Z_i\}_{i=1}^n \sim \mu^{\otimes n}$ be a labeled source-domain sample and $S'_{X'} = \{X'_j\}_{j=1}^m \sim P_{X'}^{\otimes m}$ be an unlabelled target-domain sample. The objective of UDA is to design an algorithm $\mathcal{A}$ that takes $S$ and $S'_{X'}$ as the input and outputs a weight $W \in \mathcal{W}$, giving rise to a predictor $f_W \in \mathcal{F}$ that "works well" on the target domain. Note that the algorithm $\mathcal{A}$ is characterized by a conditional distribution $P_{W|S,S'_{X'}}$.

Let $\ell : \mathcal{Y} \times \mathcal{Y} \to \mathbb{R}_0^+$ be a loss function. The population risk for each $w \in \mathcal{W}$ in the target domain is defined as
$$R_{\mu'}(w) \triangleq \mathbb{E}_{Z'}[\ell(f_w(X'), Y')]$$
and a good UDA algorithm hopes to return a weight $w$ that minimizes this risk. Since $\mu'$ is unknown, one often uses recourse to the empirical risk in the source domain, defined as
$$R_S(w) \triangleq \frac{1}{n} \sum_{i=1}^n \ell(f_w(X_i), Y_i).$$

Generalization error in this setting measures how well the hypothesis returned from the algorithm generalizes from the source-domain training sample to the target-domain unknown distribution $\mu'$. Taking into account the stochastic nature of the algorithm $\mathcal{A}$, a natural notion of generalization error for UDA can be defined by
$$\mathrm{Err} \triangleq \mathbb{E}_{W,S}[R_{\mu'}(W) - R_S(W)] = \mathbb{E}_{W,S,S'_{X'}}[R_{\mu'}(W) - R_S(W)], \tag{1}$$
where the expectation in the first expression is taken over the joint distribution of $(W, S) \sim P_{W|S} \times \mu^{\otimes n}$, and the expectation of the second expression is taken over the joint distribution of $(W, S, S'_{X'}) \sim P_{W|S,S'_{X'}} \times \mu^{\otimes n} \times P_{X'}^{\otimes m}$.

There is another notion of generalization error, more traditional in the domain adaptation literature, defined as the gap between the population risk in the target domain and that in the source domain:
$$\widetilde{\mathrm{Err}}(w) \triangleq R_{\mu'}(w) - R_\mu(w). \tag{2}$$
where $R_\mu(w) \triangleq \mathbb{E}_Z[\ell(f_w(X), Y)]$. It is apparent that $\widetilde{\mathrm{Err}}(w)$ and $\mathrm{Err}$ are related by the following triangle inequality:
$$|R_{\mu'}(w) - R_S(w)| \le |R_{\mu'}(w) - R_\mu(w)| + |R_\mu(w) - R_S(w)|.$$
where the second term on the right hand side is the standard generalization error in the source domain, which can be bounded by classical learning-theoretic tools, e.g., Rademacher complexity (Bartlett & Mendelson, 2002). Thus, bounding $\widetilde{\mathrm{Err}}(w)$ helps bounding $\mathrm{Err}$.

This paper studies both notions of generalization error for UDA. Specifically, starting from Section 5, we will mainly use information-theoretic tools to bound $\mathrm{Err}$ directly, without going through $\widetilde{\mathrm{Err}}(w)$. For the ease of reference, we refer to $\widetilde{\mathrm{Err}}(w)$ as the *population-to-population (PP) generalization error for $w$* and $\mathrm{Err}$ as the *expected empirical-to-population (EP) generalization error*.

The following definitions are useful.

**Definition 3.1** (Disintegrated Mutual Information). *Let $X$, $Y$ and $Z$ be random variables and $z$ be a realization of $Z$. The disintegrated mutual information of $X$ and $Y$ given $Z = z$ is $I^z(X; Y) \triangleq \mathrm{D}_{\mathrm{KL}}(P_{X,Y|Z=z}||P_{X|Z=z}P_{Y|Z=z})$.*

Note that the conditional mutual information $I(X; Y|Z) = \mathbb{E}_Z I^Z(X; Y)$.

**Definition 3.2** (Lautum Information (Palomar & Verdú, 2008)). *The lautum information between $X$ and $Y$ is defined as $L(X; Y) \triangleq \mathrm{D}_{\mathrm{KL}}(P_X P_Y||P_{XY})$.*

## 4 UPPER BOUNDS FOR PP GENERALIZATION ERROR

We now present some upper bounds for $\widetilde{\mathrm{Err}}(w)$. The key techniques used in developing these bounds are the information-theoretic tools in the style of Lemma A.1. These bounds adopt certain

KL divergence to measure the discrepancy between the source and target domains. Notably, some previously established bounds are recovered under weaker conditions. Additionally, we demonstrate that under certain conditions, the KL-based bound is an upper bound of several other discrepancy measures and hence minimizing the KL divergence forces the minimization of these other measures.

We first list some common assumptions on the loss function, which we consider in this paper.

**Assumption 1** (Boundedness). $\ell(\cdot, \cdot)$ is bounded in $[0, M]$.

**Assumption 2** (Subgaussianity). $\ell(f_w(X), Y)$ is $R$-subgaussian[1] under $\mu$ for any $w \in \mathcal{W}$.

**Remark 4.1.** *Note that Assumption 1 implies Assumption 2, i.e., if $\ell(f_w(X), Y)$ is bounded in $[0, M]$, then it is also $M/2$-subgaussian. Thus, Assumption 2 is weaker than Assumption 1.*

**Assumption 3** (Lipschitzness). $\ell(f_w(X), Y)$ is $\beta$-Lipschitz continuous in $\mathcal{Z}$ with respect to a metric $d$ on $\mathcal{Z}$ for any $w \in \mathcal{W}$, i.e., $|\ell(f_w(x_1), y_1) - \ell(f_w(x_2), y_2)| \le \beta d(z_1, z_2)$ for some metric $d$ on $\mathcal{Z}$.

**Remark 4.2.** *Note that Assumption 1 implies Assumption 3 when $d(z_1, z_2) = \mathbb{1}_{z_1 \neq z_2}$, i.e., if $\ell(f_w(X), Y)$ is bounded in $[0, M]$, then it is also $M$-Lipschitz under the discrete metric.*

**Assumption 4** (Triangle and Symmetric). $\ell(\cdot, \cdot)$ *satisfies the following:* $\ell(y_1, y_2) = \ell(y_2, y_1)$ *and* $\ell(y_1, y_2) \le \ell(y_1, y_3) + \ell(y_3, y_2)$ *for any* $y_1, y_2, y_3 \in \mathcal{Y}$.

### 4.1 Generalization Bounds via the Subgaussian Condition

The following generalization bound is established by combining Lemma A.1 and Assumption 2, and its corresponding sample complexity bound is discussed in Appendix B.8.

**Theorem 4.1.** *If Assumption 2 holds, then for any $w \in \mathcal{W}$, $\left|\widetilde{\mathrm{Err}}(w)\right| \le \sqrt{2R^2 \mathrm{D}_{\mathrm{KL}}(\mu' || \mu)}$.*

Notably this result can be turned into a generalization upper bound providing guidance to algorithm design, and at the same time it provides a lower bound of the generalization error, highlighting some fundamental difficulty of the learning task. To illustrate this, we present a corollary while noting that similar development can also be applied to other bounds presented later in this paper.

Consider that each $f_w$ is expressed as the composition $g \circ h$, where $h$ is a function mapping $\mathcal{X}$ to a representation space $\mathcal{T}$ and $g$ is a function mapping $\mathcal{T}$ to $\mathcal{Y}$. For any given $h : \mathcal{X} \to \mathcal{T}$, denote by $\mu_h$ the distribution on $\mathcal{T} \times \mathcal{Y}$ obtained by pushing forward $\mu$ via $h$, that is, $\mu_h(t, y) = \int \delta(t - h(x)) d\mu(x, y)$, where $\delta$ is the Dirac measure on $\mathcal{T}$. Similarly, let $\mu'_h$ denote the distribution on $\mathcal{T} \times \mathcal{Y}$ obtained by pushing forward $\mu'$ via $h$.

**Corollary 4.1.** *Suppose that $f_w = g \circ h$ and that Assumption 2 holds, then for any $w \in \mathcal{W}$,*

$$R_\mu(w) - \sqrt{2R^2 \mathrm{D}_{\mathrm{KL}}(\mu' || \mu)} \le R_{\mu'}(w) \le R_\mu(w) + \sqrt{2R^2 \mathrm{D}_{\mathrm{KL}}(\mu'_{\mathrm{h}} || \mu_{\mathrm{h}})}.$$

In this result, the lower bound of $R_{\mu'}(w)$ indicates a fundamental difficulty in UDA learning in that, using the same predictor mapping $f_w$, there is no way for the population risk in the target domain to be lower than that of the source domain less than a constant which depends only on the domain difference. On the other hand, the upper bound suggests that it is possible to squeeze the gap between the two population risks by choosing an appropriate representation map $h$ - evidently such a map should be attempting to align $\mu'_h$ with $\mu_h$ or to align their respective proxies.

It is also noteworthy that under Assumption 1 and due to Remark 4.1, Theorem 4.1 implies

$$\left|\widetilde{\mathrm{Err}}(w)\right| \le \frac{M}{\sqrt{2}} \sqrt{\mathrm{D}_{\mathrm{KL}}(P_{X'} || P_X) + \mathrm{D}_{\mathrm{KL}}(P_{Y'|X'} || P_{Y|X})}. \tag{3}$$

Similarly applying this result in the representation space $\mathcal{T}$, we see that Eq. (3) recovers the bound in Proposition 1 of Nguyen et al. (2022). Notice that unlike Nguyen et al. (2022), Theorem 4.1 ( or Eq. (3)) does not require the loss to be the cross-entropy loss.

Theorem 4.1 and Nguyen et al. (2022) both use the KL divergence from source domain to target domain, $\mathrm{D}_{\mathrm{KL}}(\mu' || \mu)$, and in fact, $\left|\widetilde{\mathrm{Err}}(w)\right|$ can also be upper bounded by $\mathrm{D}_{\mathrm{KL}}(\mu || \mu')$. This can be done by invoking the subgaussianality of $\ell(f_w(X'), Y')$ (rather than $\ell(f_w(X), Y)$); for bounded loss, the subgaussianality of $\ell(f_w(X'), Y')$ is also satisfied. Then we obtain the following corollary.

---

[1]A random variable $X$ is $R$-subgaussian if for any $\rho$, $\log \mathbb{E} \exp(\rho(X - \mathbb{E}X)) \le \rho^2 R^2 / 2$.

**Corollary 4.2.** *If Assumption 1 holds,* $\left|\widetilde{\mathrm{Err}}(w)\right| \leq \frac{M}{\sqrt{2}}\sqrt{\min\{\mathrm{D_{KL}}(\mu||\mu'), \mathrm{D_{KL}}(\mu'||\mu)\}} \leq$ $\frac{M}{2}\sqrt{\mathrm{D_{KL}}(\mu||\mu') + \mathrm{D_{KL}}(\mu'||\mu)}$.

**Remark 4.3.** *In the second inequality of Corollary 4.2,* $\mathrm{D_{KL}}(\mu||\mu') + \mathrm{D_{KL}}(\mu'||\mu)$ *is known as the symmetrized KL divergence, or Jeffrey's divergence (Jeffreys, 1946), and in fact, Nguyen et al. (2022) penalizes this measure between the source and target distributions in the representation space. Notice that bounds in Shui et al. (2020) are based on the JS divergence. Since there is a sharp upper bound of the JS divergence based on Jeffrey's divergence (Crooks, 2008), minimizing Jeffrey's divergence (in the representation space) will simultaneously penalize the JS divergence.*

In UDA, since $Y'$ is completely unavailable to the algorithm $\mathcal{A}$, it is impossible to minimize the misalignment of conditional distributions, i.e. $\mathrm{D_{KL}}(P_{Y'|T'}||P_{Y|T})$ where $T$ and $T'$ are representations of source domain and target domain, respectively. A common method is to assign pseudo labels to target data based on a learned source classifier (Liang et al., 2020). However, it may also cause additional issues (Shen et al., 2022). For concreteness, suppose the trained model $Q$ can well approximate the real mapping between $X$ and $Y$ on source domain (i.e. $Q_{Y|T} = P_{Y|T}$), which is usually the training objective. Let $\hat{Y}'$ be the pseudo label of $T'$ generated by the trained model, i.e., $Q_{\hat{Y}'|T'} = Q_{Y|T}$. Let $Q_{T',\hat{Y}'} = P_{T'}Q_{\hat{Y}'|T'}$, then the following holds,

$$\mathrm{D_{KL}}(P_{T',Y'}||P_{T,Y}) = \mathbb{E}_{P_{T',Y'}} \log \frac{P_{T',Y'}Q_{T',\hat{Y}'}}{Q_{T',\hat{Y}'}P_{T,Y}} = \mathrm{D_{KL}}(P_{T'}||P_T) + \mathrm{D_{KL}}(P_{Y'|T'}||Q_{\hat{Y}'|T'}). \quad (4)$$

For a specific $t'$, if $P(Y' = y'|T' = t') \neq 0$ and $Q(\hat{Y}' = y'|T' = t') = 0$, then the second term in RHS of Eq. (4), $\mathrm{D_{KL}}(P_{Y'|T'}||Q_{\hat{Y}'|T'}) \to \infty$. In this case, even when the marginal distributions are perfectly aligned, the overall value of the upper bound is large. Thus, incorrect pseudo labels may even have negative impact on the target domain performance.

In fact, the misalignment of the conditional distributions appears to be the main difficulty of UDA (Ben-David et al., 2006; Acuna et al., 2021). The next corollary suggests that this difficulty may be alleviated when the loss function satisfies the triangle property, namely, Assumption 4. It can be verified that this assumption is satisfied by the 0-1 loss [2]; this assumption has also been considered in previous works (Mansour et al., 2009; Shen et al., 2018).

**Theorem 4.2.** *If Assumption 4 holds and let* $\ell(f_{w'}(X), f_w(X))$ *be R-subgaussian for any* $w, w' \in \mathcal{W}$. *Then for any* $w$, $\widetilde{\mathrm{Err}}(w) \leq \sqrt{2R^2\mathrm{D_{KL}}(P_{X'}||P_X)} + \lambda^*$, *where* $\lambda^* = \min_{w \in \mathcal{W}} R_{\mu'}(w) + R_\mu(w)$.

Here $\lambda^*$ measures the possibility of whether the domain adaptation algorithm will succeed under the oracle knowledge of $\mu$ and $\mu'$. In particular, if the hypothesis space is large enough, the minimizer $w^*$ for the "joint population risk" $R_{\mu'}(w) + R_\mu(w)$ may give rise to $R_{\mu'}(w^*) = R_\mu(w^*) = 0$, then we're likely to generalize well on the target domain. Then the KL divergence $\mathrm{D_{KL}}(P_{X'}||P_X)$ between the two $\mathcal{X}$-marginals alone bounds the PP generalization error uniformly for all $w \in \mathcal{W}$.

This theorem motivates the strategy of penalizing $\mathrm{D_{KL}}(P_{T'}||P_T)$ in the representation space for UDA. The next theorem suggests that such an approach also penalizes other notions of domain discrepancy, for example, the key quantity in the PAC-Bayes type of domain adaptation generalization bounds (Germain et al., 2020), that is defined as

$$\mathrm{dis}(P_X, P_{X'}) \triangleq |\mathbb{E}_{W,W'}\left[\mathbb{E}_{X'}\left[\ell(f_W(X'), f_{W'}(X'))\right]\right] - \mathbb{E}_{W,W'}\left[\mathbb{E}_X\left[\ell(f_W(X), f_{W'}(X))\right]\right]|. \quad (5)$$

**Theorem 4.3.** *If* $\ell(f_{w'}(X), f_w(X))$ *is R-subgaussian for any* $f_w, f'_w \in \mathcal{F}$, *then* $\mathrm{dis}(P_X, P_{X'}) \leq \sqrt{2R^2\mathrm{D_{KL}}(P_{X'}||P_X)}$.

Note that unlike Germain et al. (2020), here we do not require the loss function to be the 0-1 loss.

## 4.2 GENERALIZATION BOUNDS VIA THE LIPSCHITZ CONDITION

We now present such generalization bound for UDA under the Lipschitz continuity assumption of the loss function, where $\mathbb{W}(\cdot, \cdot)$ denotes the Wasserstein distance.

**Theorem 4.4.** *If Assumption 3 holds, then* $\left|\widetilde{\mathrm{Err}}(w)\right| \leq \beta\mathbb{W}(\mu', \mu)$.

---

[2] Some losses that only satisfy a general version of Assumption 4 are discussed in Appendix B.10

Theorem 4.4 can be related to the KL-based bounds in the previous section when the Wasserstein distance is defined with respect to the discrete metric $d$. In this case and under bounded loss function, which is also Liptschitz continuous, Theorem 4.4 follows. On the other hand, Wasserstein distance is also equivalent to the total variation in this case, while the latter is connected to the KL divergence via Pinsker's inequality (Polyanskiy & Wu, 2019, Theorem 6.5) and the Bretagnolle-Huber inequality (Bretagnolle & Huber, 1979, Lemma 2.1). Thus, we arrive at the following result.

**Corollary 4.3.** *If Assumption 1 holds holds and let $d$ be the discrete metric, then*

$$\left| \widetilde{\mathrm{Err}}(w) \right| \le M \mathrm{TV}(\mu', \mu) \le M \sqrt{\min \left\{ \frac{1}{2} \mathrm{D_{KL}}(\mu'||\mu), 1 - e^{-\mathrm{D_{KL}}(\mu'||\mu)} \right\}}.$$

Note that results here are inspired by the work of Rodríguez Gálvez et al. (2021). Corollary 4.3 provides a tighter bound than the one in Eq. (3), as can be directly verified.

Parallel to Theorem 4.2, if the loss function satisfies the triangle property, we may establish the bound below, which recovers a similar result in (Shen et al., 2018, Theorem 1.) but without restricting the task to be binary classification or requiring the loss to be the $L_1$ distance.

**Theorem 4.5.** *If Assumption 4 holds and $\ell(f_w(X), f_{w'}(X))$ is $\beta$-Lipschitz in $\mathcal{X}$ for any $w, w' \in \mathcal{W}$, then for any $w \in \mathcal{W}$, $\widetilde{\mathrm{Err}}(w) \le \beta \mathbb{W}(P_{X'}, P_X) + \lambda^*$, where $\lambda^* = \min_{w \in \mathcal{W}} R_{\mu'}(w) + R_\mu(w)$.*

These results justify the strategy of minimizing domain discrepancy in the representation space. Since the KL-based bounds upper-bound those based on other measures of domain differences, penalizing the KL divergence will also penalize those other measures. This is practically advantageous since it is usually easier and more stable to minimize the KL divergence (Nguyen et al., 2022).

## 5 UPPER BOUNDS FOR EP GENERALIZATION ERROR AND APPLICATIONS

There are two limitations in the bounds on the PP generalization error developed so far and in the traditional analysis of UDA. First, such bounds are independent of $w$ and hence algorithm-independent. Second, although these bounds may inspire strategies to exploit the unlabelled target sample, e.g., aligning the source and target distributions in the representation space, they only provide very limited knowledge on the role that the unlabelled target sample plays. Inspired by the works of Negrea et al. (2019) and Rodríguez-Gálvez et al. (2021), we derive upper bounds for the EP generalization error that take better advantage of the dependence of the algorithm's output on the unlabelled target data. Applications of these bounds in designing the learning algorithms are also presented.

### 5.1 EP GENERALIZATION BOUNDS

**Theorem 5.1.** *Assume $\ell(f_W(X'), Y')$ is $R$-subgaussian under $P_{W, Z'|X'_j = x'_j}$ for any $x'_j \in \mathcal{X}$, then*

$$|\mathrm{Err}| \le \frac{1}{nm} \sum_{j=1}^{m} \sum_{i=1}^{n} \mathbb{E}_{X'_j} \sqrt{2R^2 I^{X'_j}(W; Z_i)} + \sqrt{2R^2 \mathrm{D_{KL}}(\mu||\mu')}.$$

**Remark 5.1.** *It is worth noting that the unlabelled target data contributes to the first term of the bound. Increasing the amount of source and target data will result in a reduction of the first term in the bound. Specifically, moving the expectation inside the square root function by Jensen's inequality and since $Z_i \perp\!\!\!\perp X'_j$, the equations $I(W; Z_i|X'_j) = I(W; Z_i|X'_j) + I(Z_i; X'_j) = I(W; Z_i) + I(X'_j; Z_i|W)$ hold by the chain rule. The term $I(W; Z_i)$ will vanish as $n \to \infty$ and the term $I(X'_j; Z_i|W)$ will also vanish as $n, m \to \infty$.*

The theorem can be turned into a version that is more practically relevant, in which the KL term is replaced with their representation-space counter-part (following a similar argument used for deriving Corollary 4.1). In addition, note that although larger sample sizes allow better estimation of that KL term, utilizing pseudo-labels for estimation may have a negative impact (as discussed in Section 4), which can be amplified by the larger sample size.

**Corollary 5.1.** *Let Assumption 1 hold. Then*

$$|\mathrm{Err}| \le \frac{M}{\sqrt{2}nm} \sum_{j=1}^{m} \sum_{i=1}^{n} \mathbb{E}_{X'_j} \sqrt{\min \left\{ I^{X'_j}(W; Z_i), L^{X'_j}(W; Z_i) \right\}} + \frac{M}{\sqrt{2}} \sqrt{\min \left\{ \mathrm{D_{KL}}(\mu||\mu'), \mathrm{D_{KL}}(\mu'||\mu) \right\}}.$$

**Theorem 5.2.** *Assume $\ell$ is Lipschitz for both $w \in \mathcal{W}$ and $z \in \mathcal{Z}$, i.e., $|\ell(f_w(x), y) - \ell(f_w(x'), y')| \leq \beta d_1(z, z')$ for all $z, z' \in \mathcal{Z}$ and $|\ell(f_w(x), y) - \ell(f_{w'}(x), y)| \leq \beta' d_2(w, w')$ for all $w, w' \in \mathcal{W}$, then*

$$|\mathrm{Err}| \leq \frac{\beta'}{nm} \sum_{j=1}^{m} \sum_{i=1}^{n} \mathbb{E}_{X'_j, Z_i} \mathbb{W}(P_{W|Z_i, X'_j}, P_{W|X'_j}) + \beta \mathbb{W}(\mu, \mu').$$

This bound is tighter than the bound in Theorem 5.1, as can be indicated by the following corollary.

**Corollary 5.2.** *Let Assumption 1 hold. Then*

$$\left| \widetilde{\mathrm{Err}} \right| \leq \frac{M}{nm} \sum_{j=1}^{m} \sum_{i=1}^{n} \mathbb{E}_{X'_j, Z_i} \left[ \mathrm{TV}(P_{W|Z_i, X'_j}, P_{W|X'_j}) \right] + M\mathrm{TV}(\mu, \mu')$$

$$\leq \frac{1}{nm} \sum_{j=1}^{m} \sum_{i=1}^{n} \mathbb{E}_{X'_j, Z_i} \sqrt{\frac{M^2}{2} \mathrm{D}_{\mathrm{KL}}(P_{W|Z_i, X'_j} || P_{W|X'_j})} + \sqrt{\frac{M^2}{2} \mathrm{D}_{\mathrm{KL}}(\mu || \mu')}.$$

Notice that to recover Theorem 5.1 from Corollary 5.2 (under Assumption 1), we can use Jensen's inequality to move the expectation over $Z_i$ to inside the square root function.

## 5.2 GRADIENT PENALTY AS AN UNIVERSAL REGULARIZER

The algorithm-dependent bound in Theorem 5.1 tells us that one can reduce the EP error by limiting the disintegrated mutual information $I^{X'_j}(W; Z_i)$. In the stochastic gradient based optimization algorithms, this term can be controlled by penalizing the gradient norm. To see this, we now consider a "noisy" iterative algorithm for updating $W$, e.g., SGLD. At each time step $t$, let the labelled mini-batch from the source domain be $Z_{B_t}$, let the unlabelled mini-batch from the target domain be $X'_{B_t}$, and let $g(W_{t-1}, Z_{B_t}, X'_{B_t})$ be the gradient at time $t$. Thus, the updating rule of $W$ is $W_t = W_{t-1} - \eta_t g(W_{t-1}, Z_{B_t}, X'_{B_t}) + N_t$ where $\eta_t$ is the learning rate and $N_t \sim \mathcal{N}(0, \sigma^2 \mathrm{I}_d)$ is an isotropic Gaussian noise. Inspired by Pensia et al. (2018), we have the following bound.

**Theorem 5.3.** *Let the total iteration number be $T$ and let $G_t = g(W_{t-1}, Z_{B_t}, X'_{B_t})$, then*

$$|\mathrm{Err}| \leq \sqrt{\frac{R^2}{n} \sum_{t=1}^{T} \frac{\eta_t^2}{\sigma_t^2} \mathbb{E}_{S'_{X'}, W_{t-1}, S} \left[ \left|\left| G_t - \mathbb{E}_{Z_{B_t}}[G_t] \right|\right|^2 \right]} + \sqrt{2R^2 \mathrm{D}_{\mathrm{KL}}(\mu || \mu')}.$$

**Remark 5.2.** *Considering a noisy iterative algorithm here is merely for simplifying analysis. In fact, it is also possible to analyze the original iterative gradient optimization method without noise injected. For example, one can follow the same development in (Neu et al., 2021; Wang & Mao, 2022a) to analyze vanilla SGD. In that case, there will be some residual terms in the bound.*

Theorem 5.3 hints that to reduce the generalization error, one can simply restrict the gradient norm at each step (so that $||G_t - \mathbb{E}_{Z_{B_t}}[G_t]||^2$ is reduced). This strategy will also restrict the distance between the final output $W_T$ and the initialization $W_0$, effectively shrinking the hypothesis space accessible by the algorithm. We also note that the importance of gradient penalty has been theoretically justified in the supervised learning setting (Negrea et al., 2019; Haghifam et al., 2020; Smith et al., 2021; Rodríguez-Gálvez et al., 2021; Neu et al., 2021; Wang & Mao, 2022a;b).

Indeed, adding gradient penalty can be applied to any existing UDA algorithm and it is simple but effective in practice. Later on we will show that even when the algorithm $\mathcal{A}$ does not access to any target data, in which case $I(W; Z_i | X'_j)$ reduces to $I(W; Z_i)$ and $g(W_{t-1}, Z_{B_t}, X'_{B_t})$ becomes $g(W_{t-1}, Z_{B_t})$, minimizing the empirical loss of source domain sample while penalizing gradient norm will still improve the performance. Notice that gradient penalty has been used in standard supervised learning as a regularization technique (Geiping et al., 2022; Jastrzebski et al., 2021). It is also used in Wasserstein distance based adversarial adaptation (Gulrajani et al., 2017; Shen et al., 2018), and their motivation is to stabilize the training to avoid gradient vanishing problem. Here we suggest, with strong theoretical justification, that gradient penalty is a universal technique for improving the generalization performance in UDA for any gradient-based learning method.

Notably the bound in Theorem 5.3 only depends on the size $n$ of labelled source sample and does not explicitly depend on $m$, the size of unlabelled target sample. With a more careful design, if we

consider the mutual information as the expected KL divergence of a posterior and a prior, based on $I^{X'_j}(W; Z_i)$ in Theorem 5.1, it is possible to create a target-data-dependent prior and derive a tighter bound based on some quantity similar to "gradient incoherence" in Negrea et al. (2019).

### 5.3 Controlling Label Information for KL Guided Marginal Alignment

Consider instances in the representation space, $Z = (T, Y)$ and $Z' = (T', Y)$. Theorem 5.1 also encourages us to align the distributions of two domains in the representation space, as argued earlier. Then the KL guided marginal alignment algorithm proposed in Nguyen et al. (2022) can be invoked here. One may notice that Theorem 5.1 uses $D_{KL}(\mu||\mu')$ while Nguyen et al. (2022) uses $D_{KL}(\mu'||\mu)$. As already discussed in Section 4, this inconsistency can be ignored when the loss is bounded (see Corollary 5.1).

Most domain adaptation algorithms aim to align the marginal distributions of two domains in the representation space. However, without accessing to $Y'$, it remains unknown if an UDA algorithm will work well since we cannot guarantee that discrepancy between conditional distribution $P_{Y|T}$ and $P_{Y'|T'}$ won't become too large when we align the marginals. In Nguyen et al. (2022), the authors show that $D_{KL}(P_{Y'|T'}||P_{Y|T})$ can be upper-bounded by $D_{KL}(P_{Y'|X'}||P_{Y|X})$, if $I(X; Y) = I(T; Y)$. The authors then argue that penalizing the KL divergence of the marginals is safe.

We now argue that in practice the condition $I(X; Y) = I(T; Y)$ can be difficult to satisfy if the cross-entropy loss is used to define the source-domain empirical risk.

By data processing inequality on $Y - X - T$, we know that $I(X; Y) \geq I(T; Y) = H(Y) - H(Y|T)$. Thus, to let $I(T; Y)$ reach its maximum, one must minimize $H(Y|T)$. On the other hand, let $Q_{Y|T,W}$ be the predictive distribution of labels in the source domain generated by the classifier. The expected cross-entropy loss for each $Z_i$ in the representation space is then

$$\mathbb{E}_{W,Z_i}[\ell(f_W(T_i), Y_i)] = \mathbb{E}_{Z_i}\left[\mathbb{E}_{W|Z_i}\left[-\log Q_{Y_i|T_i,W}\right]\right],$$

which also decomposes as (Achille & Soatto, 2018; Harutyunyan et al., 2020)

$$\mathbb{E}_{W,Z_i}[\ell(f_W(T_i), Y_i)] = H(Y_i|T_i) + \mathbb{E}_{T_i,W}\left[D_{KL}(P_{Y_i|T_i,W}||Q_{Y_i|T_i,W})\right] - I(W; Y_i|T_i). \quad (6)$$

Then minimizing the expected cross-entropy loss may not adequately reduce $H(Y_i|T_i)$ but rather cause $I(W; Y_i|T_i)$ to significantly increase, particularly when the model capacity is large. This may have two negative effects. First, the condition $I(X; Y) = I(T; Y)$ is significantly violated, and $D_{KL}(P_{Y'|T'}||P_{Y|T})$ is no longer upper bounded by $D_{KL}(P_{Y'|X'}||P_{Y|X})$. Hence, aligning the two marginals alone may not be adequate. Second, large $I(W; Y_i|T_i)$ indicates $W$ just simply memorizes the label $Y_i$, resulting a form of overfitting and hurting the generalization performance.

The key take-away from the above analysis is that when aligning the marginals in UDA, controlling the source label information in the weights can be important to achieve good cross-domain generalization. A similar message can also be deduced from Theorem 5.1, when it is viewed in the representation space and noting $I^{T'_j}(W; Z_i) = I^{T'_j}(W; T_i) + I^{T'_j}(W; Y_i|T_i)$.

To control label information, Harutyunyan et al. (2020) proposed an approach called LIMIT. However, this method is rather complicated and arguably hard to train in domain adaptation (see Appendix C.8). We now derive a simple alternative strategy for this purpose.

Notice that $I^{T'_j}(W; Y_i|T_i) \leq \inf_Q \mathbb{E}_{T_i}\left[D_{KL}(P_{W|Y_i,T_i,T'_j=t'_j}||Q_{W|T_i,T'_j=t'_j})\right]$, which is a simple extension of variational representation of mutual information (Polyanskiy & Wu, 2019, Corollary 3.1.). Here $Q$ could be any distribution. By assuming $P = \mathcal{N}(W, \sigma^2 I_d|Y_i, T_i, T'_j = t'_j)$ and taking $Q = \mathcal{N}(\widetilde{W}, \tilde{\sigma}^2 I_d|T_i, T'_j = t'_j)$, we have

$$I^{T'_j}(W; Y_i|T_i) \leq \inf_Q \mathbb{E}_{T_i}\left[D_{KL}(P_{W|Y_i,T_i,T'_j=t'_j}||Q_{\tilde{W}|T_i,T'_j=t'_j})\right] \propto ||W - \widetilde{W}||^2.$$

Thus, we may create an auxiliary classifier $f_{\tilde{w}}$ that is not allowed to access to the real source label $Y$. In each iteration, we use the pseudo labels of target data (and source data) assigned by $f_w$ to train $f_{\tilde{w}}$ and adding $||W - \widetilde{W}||^2$ as a regularizer in the training of $W$. The algorithm is given in the Appendix. Remarkably the regularizer here resembles "Projection Norm" designed in Yu et al. (2022) for out-of-distribution generalization.

Table 1: RotatedMNIST and Digits. Results of baselines are reported from Nguyen et al. (2022).

| Method | RotatedMNIST ($0°$ as source domain) | | | | | | Digits | | | |
| | $15°$ | $30°$ | $45°$ | $60°$ | $75°$ | Ave | $M \rightarrow U$ | $U \rightarrow M$ | $S \rightarrow M$ | Ave |
|---|---|---|---|---|---|---|---|---|---|---|
| ERM | 97.5±0.2 | 84.1±0.8 | 53.9±0.7 | 34.2±0.4 | 22.3±0.5 | 58.4 | 73.1±4.2 | 54.8±6.2 | 65.9±1.4 | 64.6 |
| DANN | 97.3±0.4 | 90.6±1.1 | 68.7±4.2 | 30.8±0.6 | 19.0±0.6 | 61.3 | 90.7±0.4 | 91.2±0.8 | 71.1±0.5 | 84.3 |
| MMD | 97.5±0.1 | 95.3±0.4 | 73.6±2.1 | 44.2±1.8 | 32.1±2.1 | 68.6 | 91.8±0.3 | 94.4±0.5 | 82.8±0.3 | 89.7 |
| CORAL | 97.1±0.3 | 82.3±0.3 | 56.0±2.4 | 30.8±0.2 | 27.1±1.7 | 58.7 | 88.0±1.9 | 83.3±0.1 | 69.3±0.6 | 80.2 |
| WD | 96.7±0.3 | 93.1±1.2 | 64.1±3.3 | 41.4±7.6 | 27.6±2.0 | 64.6 | 88.2±0.6 | 60.2±1.8 | 68.4±2.5 | 72.3 |
| KL | 97.8±0.1 | 97.1±0.2 | 93.4±0.8 | 75.5±2.4 | 68.1±1.8 | 86.4 | 98.2±0.2 | 97.3±0.5 | 92.5±0.9 | 96.0 |
| ERM-GP | 97.5±0.1 | 86.2±0.5 | 62.0±1.9 | 34.8±2.1 | 26.1±1.2 | 61.2 | 91.3±1.6 | 72.7±4.2 | 68.4±0.2 | 77.5 |
| KL-GP | 98.2±0.2 | 96.9±0.1 | 95.0±0.6 | **88.0±8.1** | **78.1±2.5** | **91.2** | 98.8±0.1 | **97.8±0.1** | **93.8±1.1** | **96.8** |
| KL-CL | **98.4±0.2** | **97.3±0.2** | **95.6±0.1** | 83.0±8.2 | 73.6±4.0 | 89.6 | **98.9±0.1** | 97.7±0.1 | 93.0±0.3 | 96.5 |

## 6 EXPERIMENTAL RESULTS

We perform experiments to verify the proposed techniques inspired by our theory.

**Datasets** We select two popular small datasets, RotatedMNIST and Digits, to compare the different methods. RotatedMNIST is built based on the MNIST dataset (LeCun et al., 2010) and consists of six domains, each containing $11,666$ images. These six domains are rotated MNIST images with rotation angle $0°, 15°, 30°, 45°, 60°$ and $75°$, respectively. We will take the original MNIST dataset ($0°$) as the source domain and take other five domains as target domains. Hence, there are five domain adaptation tasks on RotatedMNIST. Digits consists of three sub-datasets, namely MNIST, USPS (Hull, 1994) and SVHN (Netzer et al., 2011), and the corresponding domain adaptation tasks are MNIST→USPS (**M**→**U**), USPS→MNIST (**U**→**M**), SVHN→MNIST (**S**→**M**).

**Compared Methods** Baseline methods are some popular marginal alignment UDA methods including **DANN** (Ganin et al., 2016), **MMD** (Li et al., 2018), **CORAL** (Sun & Saenko, 2016), **WD** (Shen et al., 2018) and **KL** (Nguyen et al., 2022). We also choose **ERM** as another baseline, in which only the source-domain sample is accessible during training. To verify the strategies inspired by our theory, we first add the gradient penalty to the ERM algorithm (**ERM-GP**), and we then combine gradient penalty (GP) and controlling label information (CL) with the recent proposed KL guided marginal alignment method, which are denoted by **KL-GP** and **KL-CL**, respectively.

**Implementation Details** Most of our implementation is based on the *DomainBed* suite (Gulrajani & Lopez-Paz, 2021). Other settings exactly follow Nguyen et al. (2022) and the results of baseline methods are taken from Nguyen et al. (2022). Specifically, each algorithm is run three times and we show the average performance with the error bar. Every dataset has a validation set, and the model selection scheme is based on the best performance achieved on the validation set of target domain during training (oracle). The hype-parameter searching process is also built upon the implementation in the *DomainBed* suite. Other details and additional experiments can be found in Appendix.

**Results** From Table 1, we first notice that gradient penalty allows **ERM** to perform more comparably to other marginal alignment methods. For example, on RotatedMNIST, **ERM-GP** outperforms **CORAL** and performs nearly the same with **DANN**. On Digits, **ERM-GP** outperforms **WD**. When GP and CL combined with KL guided algorithm, we can see that the performance can be further boosted. This justifies the discussion in Section 5.2 and Section 5.3.

## 7 CONCLUSION

Despite that the numerous learning techniques have been developed for domain adaptation, significant room exists for more in-depth theoretical understanding and more principled design of learning algorithms. This paper presents the information-theoretic analysis for unsupervised domain adaptation, where we query two notions of the generalization errors in this context and present novel learning bounds. Some of these bounds recover the previous KL-based bounds under different conditions and confirm the insights in the learning algorithms that align the source and target distributions in the representation space. Our other bounds are algorithm-dependent, better exploiting the unlabelled target data, which have inspired novel and yet simple schemes for the design of learning algorithms. We demonstrate the effectiveness of these schemes on standard benchmark datasets.

ACKNOWLEDGMENTS

This work is supported partly by an NSERC Discovery grant and a National Research Council of Canada (NRC) Collaborative R&D grant (AI4D-CORE-07). Ziqiao Wang is also supported in part by the NSERC CREATE program through the Interdisciplinary Math and Artificial Intelligence (INTER-MATH-AI) project. The authors would like to thank the anonymous reviewers for their careful reading and valuable suggestions.

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

# Appendix

## Table of Contents

## A  SOME PREREQUISITE DEFINITIONS AND USEFUL LEMMAS

**Definition A.1** (Wasserstein Distance)**.** *Let $d(\cdot, \cdot)$ be a metric and let $P$ and $Q$ be probability measures on $\mathcal{X}$. Denote $\Gamma(P, Q)$ as the set of all couplings of $P$ and $Q$ (i.e. the set of all joint distributions on $\mathcal{X} \times \mathcal{X}$ with two marginals being $P$ and $Q$), then the Wasserstein Distance of order one between $P$ and $Q$ is defined as $\mathbb{W}(P, Q) \triangleq \inf_{\gamma \in \Gamma(P,Q)} \int_{\mathcal{X} \times \mathcal{X}} d(x, x') d\gamma(x, x')$.*

**Remark A.1.** *Similar to Rodríguez Gálvez et al. (2021), here we mainly focus on $1$-Wasserstein distance but all the upper bounds based on $1$-Wasserstein distance also holds for higher order Wasserstein distance by Hölder's inequality (Cédric, 2008, Remark 6.6).*

**Definition A.2** (Total Variation)**.** *The total variation between two probability measures $P$ and $Q$ is $\mathrm{TV}(P, Q) \triangleq \sup_E |P(E) - Q(E)|$, where the supremum is over all measurable set $E$.*

**Remark A.2.** *Note that the total variation equals to the Wasserstein distance under the discrete metric (or Hamming distortion) $d(x, x') = \mathbb{1}_{x \neq x'}$ where $\mathbb{1}$ is the indicator function (Cédric, 2008, Theorem 6.15).*

The key quantity in the most information-theoretic generalization bounds is the mutual information between algorithm's input and output. Specifically, the core technique behind these bounds is the well-known Donsker-Varadhan representation of KL divergence (Polyanskiy & Wu, 2019, Theorem 3.5).

**Lemma A.1** (Donsker and Varadhan's variational formula). *Let $Q$, $P$ be probability measures on $\Theta$, for any bounded measurable function $f : \Theta \to \mathbb{R}$, we have $\mathrm{D_{KL}}(Q||P) = \sup_f \mathbb{E}_{\theta \sim Q} [f(\theta)] - \log \mathbb{E}_{\theta \sim P} [\exp f(\theta)]$.*

**Remark A.3.** *Motivated by the classic $f$-divergence, Acuna et al. (2021) proposed a discrepancy measure called $\mathrm{D}_{\mathcal{H}}^{\phi}$-discrepancy (or $\mathrm{D}_{h,\mathcal{H}}^{\phi}$-discrepancy). As KL divergence belongs to the family of $f$-divergences and both Acuna et al. (2021) and our work use the variational representation of divergence, there appears to be a connection between our work (in Section 4) and theirs. However, it's important to note that the variational characterization of $f$-divergence used in Acuna et al. (2021) is based on the results of Nguyen et al. (2010), while the Donsker-Varadhan representation of KL divergence (see Lemma A.1) used in our paper cannot be directly obtained from their variational characterization (Jiao et al., 2017; Agrawal & Horel, 2020). In fact, simply choosing $x \log x$ as the conjugate function would result in a weaker bound than Lemma A.1. Therefore, while there is some similarity between our results and those of Acuna et al. (2021), our results in Section 4 cannot be directly derived from theirs.*

Similar to Xu & Raginsky (2017, Lemma 1.), we need the following lemma as a main tool.

**Lemma A.2.** *Let $Q$ and $P$ be probability measures on $\Theta$. Let $\theta' \sim Q$ and $\theta \sim P$. If $g(\theta)$ is $R$-subgaussian, then,*

$$|\mathbb{E}_{\theta' \sim Q} [g(\theta')] - \mathbb{E}_{\theta \sim P} [g(\theta)]| \leq \sqrt{2R^2 \mathrm{D_{KL}}(Q||P)}.$$

*Proof.* Let $f = t \cdot g$ for any $t \in \mathbb{R}$, by Lemma A.1, we have

$$
\begin{aligned}
\mathrm{D_{KL}}(Q||P) &\geq \sup_t \mathbb{E}_{\theta' \sim Q} [tg(\theta')] - \log \mathbb{E}_{\theta \sim P} [\exp t \cdot g(\theta)] \\
&= \sup_t \mathbb{E}_{\theta' \sim Q} [tg(\theta')] - \log \mathbb{E}_{\theta \sim P} [\exp t(g(\theta) - \mathbb{E}_{\theta \sim P} [g(\theta)] + \mathbb{E}_{\theta \sim P} [g(\theta)])] \\
&= \sup_t \mathbb{E}_{\theta' \sim Q} [tg(\theta')] - \mathbb{E}_{\theta \sim P} [tg(\theta)] - \log \mathbb{E}_{\theta \sim P} [\exp t(g(\theta) - \mathbb{E}_{\theta \sim P} [g(\theta)])] \\
&\geq \sup_t t \left( \mathbb{E}_{\theta' \sim Q} [g(\theta')] - \mathbb{E}_{\theta \sim P} [g(\theta)] \right) - t^2 R^2 / 2,
\end{aligned}
$$

where the last inequality is by the subgaussianity of $g(\theta)$.

Then consider the case of $t > 0$ and $t < 0$ ($t = 0$ is trivial), by AM–GM inequality (i.e. the arithmetic mean is greater than or equal to the geometric mean), the following is straightforward,

$$|\mathbb{E}_{\theta' \sim Q} [g(\theta')] - \mathbb{E}_{\theta \sim P} [g(\theta)]| \leq \sqrt{2R^2 \mathrm{D_{KL}}(Q||P)}.$$

This completes the proof. $\qquad\square$

The following lemma is the Kantorovich–Rubinstein duality of Wasserstein distance (Cédric, 2008).

**Lemma A.3** (KR duality). *For any two distributions $P$ and $Q$, we have*

$$\mathbb{W}(P, Q) = \sup_{f \in 1 - \mathrm{Lip}(\rho)} \int_{\mathcal{X}} f dP - \int_{\mathcal{X}} f dQ,$$

*where the supremum is taken over all $1$-Lipschitz functions in the metric $d$, i.e. $|f(x) - f(x')| \leq d(x, x')$ for any $x, x' \in \mathcal{X}$.*

To connect total variation with KL divergence , we will use Pinsker's inequality (Polyanskiy & Wu, 2019, Theorem 6.5) and Bretagnolle-Huber inequality (Bretagnolle & Huber, 1979, Lemma 2.1) in this paper, for more discussion about these two inequalities, we refer readers to Canonne (2022).

$$S'_{X'}$$
$$\downarrow$$
$$S \quad \rightarrow \quad W$$
$$\downarrow \quad \swarrow$$
$$F$$

Figure 1: The relationship between random variables in UDA, where $F = R_{\mu'}(W) - R_S(W)$.

**Lemma A.4** (Pinsker's inequality). $\mathrm{TV}(P, Q) \leq \sqrt{\frac{1}{2} \mathrm{D_{KL}}(P||Q)}$.

**Lemma A.5** (Bretagnolle-Huber inequality). $\mathrm{TV}(P, Q) \leq \sqrt{1 - e^{-\mathrm{D_{KL}}(P||Q)}}$.

Below is the variational formula (or golden formula) of mutual information.

**Lemma A.6** (Polyanskiy & Wu (2019, Corollary 3.1.)). *For two random variables $X$ and $Y$, we have*

$$I(X; Y) = \inf_P \mathbb{E}_X \left[ \mathrm{D_{KL}}(Q_{Y|X}||P) \right],$$

*where the infimum is achieved at $P = Q_Y$.*

## B    OMITTED PROOFS AND ADDITIONAL RESULTS IN SECTION 4

### B.1    PROOF OF THEOREM 4.1

*Proof.* Let $Q = \mu'$, $P = \mu$ and $g = \ell$, then Theorem 4.1 comes directly from Lemma A.2.    □

### B.2    PROOF OF COROLLARY 4.2

*Proof.* As discussed in Remark 4.1, when the loss is bounded in $[0, M]$, it is guaranteed to be $\frac{M}{2}$-subgaussian for any $w \in \mathcal{W}$. Then, similar to the proof of Theorem 4.1, let $Q = \mu$, $P = \mu'$ and $g = \ell$ and $R = \frac{M}{2}$, then the following bound holds by Lemma A.2,

$$\left| \widetilde{\mathrm{Err}}(w) \right| \leq \sqrt{\frac{M^2}{2} \mathrm{D_{KL}}(\mu||\mu')}.$$

Then, by $\min\{A, B\} \leq \frac{1}{2}(A + B)$, the remaining part is straightforward,

$$\left| \widetilde{\mathrm{Err}}(w) \right| \leq \frac{M}{\sqrt{2}} \sqrt{\min\{\mathrm{D_{KL}}(\mu||\mu'), \mathrm{D_{KL}}(\mu'||\mu)\}} \leq \frac{M}{2} \sqrt{\mathrm{D_{KL}}(\mu||\mu') + \mathrm{D_{KL}}(\mu'||\mu)}.$$

This completes the proof.    □

### B.3    PROOF OF THEOREM 4.2

*Proof.* Let $w^* = \arg\min_{w \in \mathcal{W}} \mathbb{E}_{Z'} \left[ \ell(f_w(X'), Y') \right] + \mathbb{E}_Z \left[ \ell(f_w(X), Y) \right]$. By Lemma A.1,

$$\mathrm{D_{KL}}(P_{X'}||P_X) \geq \sup_{t \in \mathbb{R}, w \in \mathcal{W}} \mathbb{E}_{X'} \left[ t\ell(f_w(X'), f_{w^*}(X')) \right] - \log \mathbb{E}_X \left[ e^{t\ell(f_w(X), f_{w^*}(X))} \right].$$

Recall that $\ell(f_{w'}(X), f_w(X))$ is $R$-subgaussian, by using Lemma A.2 (let $Q = P_{X'}$, $P = P_X$ and $g(\cdot) = \ell(f_{w'}(\cdot), f_w(\cdot))$), we have

$$|\mathbb{E}_{X'} \left[ \ell(f_w(X'), f_{w^*}(X')) \right] - \mathbb{E}_X \left[ \ell(f_w(X), f_{w^*}(X)) \right]| \leq \sqrt{2R^2 \mathrm{D_{KL}}(P_{X'}||P_X)}. \quad (7)$$

For any $f_w \in \mathcal{F}$, by the symmetric and triangle property of the loss, we have

$$\mathbb{E}_{Z'}\left[\ell(f_w(X'), Y')\right]$$

$$\leq \mathbb{E}_{X'}\left[\ell(f_w(X'), f_{w^*}(X'))\right] + \mathbb{E}_{Z'}\left[\ell(f_{w^*}(X'), Y')\right]$$

$$\leq \mathbb{E}_X\left[\ell(f_w(X), f_{w^*}(X))\right] + \sqrt{2R^2 \mathrm{D}_{\mathrm{KL}}(P_{X'}||P_X)} + \mathbb{E}_{Z'}\left[\ell(f_{w^*}(X'), Y')\right] \qquad (8)$$

$$= \int_x \ell(f_w(x), f_{w^*}(x)) dP_X(x) + \sqrt{2R^2 \mathrm{D}_{\mathrm{KL}}(P_{X'}||P_X)} + \mathbb{E}_{Z'}\left[\ell(f_{w^*}(X'), Y')\right]$$

$$= \int_x \int_y \ell(f_w(x), f_{w^*}(x)) dP_{Y|X=x}(y) dP_X(x) + \sqrt{2R^2 \mathrm{D}_{\mathrm{KL}}(P_{X'}||P_X)} + \mathbb{E}_{Z'}\left[\ell(f_{w^*}(X'), Y')\right]$$

$$\leq \int_x \int_y \ell(f_w(x), y) + \ell(y, f_{w^*}(x)) dP_{Y|X=x}(y) dP_X(x) + \sqrt{2R^2 \mathrm{D}_{\mathrm{KL}}(P_{X'}||P_X)} + \mathbb{E}_{Z'}\left[\ell(f_{w^*}(X'), Y')\right]$$

$$(9)$$

$$= \mathbb{E}_Z\left[\ell(f_w(X), Y)\right] + \mathbb{E}_Z\left[\ell(Y, f_{w^*}(X))\right] + \sqrt{2R^2 \mathrm{D}_{\mathrm{KL}}(P_{X'}||P_X)} + \mathbb{E}_{Z'}\left[\ell(f_{w^*}(X'), Y')\right],$$

where Eq. (8) is by Eq. (7) and Eq. (9) is again by the triangle property of the loss function.

Thus, $\widetilde{\mathrm{Err}}(w) \leq \sqrt{2R^2 \mathrm{D}_{\mathrm{KL}}(\mathrm{P}_{X'}||\mathrm{P}_X)} + \lambda^*$, which completes the proof. $\qquad \square$

## B.4 PROOF OF THEOREM 4.3

*Proof.* By Lemma A.1,

$$\mathrm{D}_{\mathrm{KL}}(P_{X'}||P_X) \geq \sup_{t \in \mathbb{R}, w, w' \in \mathcal{W}^2} \mathbb{E}_{X'}\left[t\ell(f_w(X'), f_{w'}(X'))\right] - \log \mathbb{E}_X\left[e^{t\ell(f_w(X), f_{w'}(X))}\right]$$

$$\geq \sup_{t \in \mathbb{R}} \mathbb{E}_{W,W'}\left[\mathbb{E}_{X'}\left[t\ell(f_W(X'), f_{W'}(X'))\right] - \log \mathbb{E}_X\left[e^{t\ell(f_W(X), f_{W'}(X))}\right]\right]$$

$$\geq \sup_{t \in \mathbb{R}} \mathbb{E}_{W,W'}\left[\mathbb{E}_{X'}\left[t\ell(f_W(X'), f_{W'}(X'))\right]\right] - \log \mathbb{E}_{W,W'}\left[\mathbb{E}_X\left[e^{t\ell(f_W(X), f_{W'}(X))}\right]\right],$$

where the last inequality is by applying Jensen's inequality to the logarithm function, which is concave.

By the subgaussian assumption,

$$\left|\mathbb{E}_{W,W'}\left[\mathbb{E}_{X'}\left[\ell(f_W(X'), f_{W'}(X'))\right]\right] - \mathbb{E}_{W,W'}\left[\mathbb{E}_X\left[\ell(f_W(X), f_{W'}(X))\right]\right]\right| \leq \sqrt{2R^2 \mathrm{D}_{\mathrm{KL}}(P_{X'}||P_X)}.$$

This concludes the proof. $\qquad \square$

## B.5 PROOF OF THEOREM 4.4

*Proof.* From the definition, we have

$$\left|\widetilde{\mathrm{Err}}(w)\right| = \left|\mathbb{E}_{Z'}\left[\ell(f_w(X'), Y')\right] - \mathbb{E}_Z\left[\ell(f_w(X), Y)\right]\right|$$

$$\leq \beta \mathbb{W}(\mu, \mu').$$

where the last inequality is by the KR duality of Wasserstein distance (see Lemma A.3). $\qquad \square$

## B.6 PROOF OF COROLLARY 4.3

*Proof.* When $d$ is the discrete metric, Wasserstein distance is equal to the total variation, then by Theorem 4.4,

$$\left|\widetilde{\mathrm{Err}}(w)\right| \leq \beta \mathrm{TV}(\mu', \mu),$$

The remaining part is by using Lemma A.4 and Lemma A.5:

$$\beta \mathrm{TV}(\mu', \mu) \leq \beta \sqrt{\min\left\{\frac{1}{2}\mathrm{D}_{\mathrm{KL}}(\mu'||\mu), 1 - e^{-\mathrm{D}_{\mathrm{KL}}(\mu'||\mu)}\right\}}.$$

Then, if $\ell$ is bounded by $M$, we can replace $\beta$ by $M$ above, which completes the proof. $\qquad \square$

## B.7 Proof of Theorem 4.5

*Proof.* Let $w^* = \arg\min_{w \in \mathcal{W}} \mathbb{E}_{Z'} [\ell(f_w(X'), Y')] + \mathbb{E}_Z [\ell(f_w(X), Y)]$.

If $\ell(f_w(X), f_{w'}(X))$ is $\beta$-Lipschitz in $\mathcal{X}$ for any $w, w' \in \mathcal{W}$, then similar to Theorem 4.4, it's easy to show that

$$\mathbb{E}_{X'} [\ell(f_w(X'), f^*(X'))] - \mathbb{E}_X [\ell(f_w(X), f^*(X))] \leq \beta \mathbb{W}(P'_X, P_X) \tag{10}$$

For any $f_w \in \mathcal{F}$, by the symmetric and triangle property of the loss, we have

$$\begin{aligned}
&\mathbb{E}_{Z'} [\ell(f_w(X'), Y')] \\
\leq &\mathbb{E}_{X'} [\ell(f_w(X'), f_{w^*}(X'))] + \mathbb{E}_{Z'} [\ell(f_{w^*}(X'), Y')] \\
\leq &\mathbb{E}_X [\ell(f_w(X), f_{w^*}(X))] + \beta \mathbb{W}(P'_X, P_X) + \mathbb{E}_{Z'} [\ell(f_{w^*}(X'), Y')] \\
\leq &\mathbb{E}_Z [\ell(f_w(X), Y)] + \mathbb{E}_Z [\ell(Y, f_{w^*}(X))] + \beta \mathbb{W}(P'_X, P_X) + \mathbb{E}_{Z'} [\ell(f_{w^*}(X'), Y')],
\end{aligned} \tag{11}$$

where Eq. (11) is by Eq. (10) and the last inequality is again by the triangle property of the loss function. This completes the proof. $\square$

## B.8 Additional Results: Sample Complexity Bounds

One of the main ingredients to derive our sample complexity bound is the following lemma, where a concentration bound for a class of unbounded functions is given.

**Lemma B.1** (Cortes et al. (2019, Corollary 9)). *Let $\kappa > 2$ and $\mathcal{G} = \{g : \mathcal{Z} \to \mathbb{R} \text{ s.t. } \mathbb{E}_\mu [e^{g(Z)}] < +\infty\}$. Assume $\mathbb{E}_\mu [g(Z)^\kappa] < +\infty$ for all $g \in \mathcal{G}$. Let $\hat{\mu}$ be the empirical distributions consist of $n$ data points sampled i.i.d. from $\mu$. If $\mathcal{G}$ has the finite pseudo-dimension $d$, then for $\forall \delta \in (0, 1)$, the following inequality holds for all $g \in \mathcal{G}$ with probability at least $1 - \delta$,*

$$\mathbb{E}_\mu [g(Z)] \leq \mathbb{E}_{\hat{\mu}} [g(Z)] + 2\Lambda(\kappa) \sqrt[\kappa]{\mathbb{E}_\mu [g(Z)^\kappa]} \sqrt{\frac{1}{n} \left( d \log \frac{2en}{d} + \log \frac{4}{\delta} \right)},$$

*where $\Lambda(\kappa) = \left(\frac{1}{2}\right)^{\frac{2}{\kappa}} \left(\frac{\kappa}{\kappa-2}\right)^{\frac{\kappa-1}{\kappa}}$.*

Below is another useful lemma for the bounded case, which comes from Mohri et al. (2018, Theorem 11.8) with a slight modification (by invoking a different VC-dimension based generalization bound from Vapnik (1998)).

**Lemma B.2.** *Let $\mathcal{F} = \{f : \mathcal{Z} \to \mathbb{R}^+\}$. Assume $\mathbb{E}_\mu [f(Z)] < M$ for all $f \in \mathcal{F}$ for some constant $M > 0$. Let $\hat{\mu}$ be the empirical distributions consist of $n$ data points sampled i.i.d. from $\mu$. If $\mathcal{F}$ has the finite pseudo-dimension $d$, then for $\forall \delta \in (0, 1)$, the following inequality holds for all $f \in \mathcal{F}$ with probability at least $1 - \delta$,*

$$\mathbb{E}_\mu [f(Z)] \leq \mathbb{E}_{\hat{\mu}} [f(Z)] + 2M \sqrt{\frac{1}{n} \left( d \log \frac{2en}{d} + \log \frac{4}{\delta} \right)}.$$

We are now in a position to state our sample complexity bound.

**Theorem B.1.** *Let $\hat{\mu}$ and $\hat{\mu}'$ be the empirical distributions consist of $n$ source data and $m$ target data sampled i.i.d. from $\mu$ and $\mu'$, respectively. Let $\mathcal{G} = \{g : \mathcal{Z} \to \mathbb{R} \text{ s.t. } \mathbb{E}_\mu [e^{g(Z)}] < \infty\}$ with finite pseudo-dimension $d_1$, and let the pseudo-dimension of $\{\exp \circ g | g \in \mathcal{G}\}$ be $d_2$. Let $\kappa > 2$ and assume that $\mathbb{E}_\mu [g(Z)^\kappa] < +\infty$ for all $g \in \mathcal{G}$. Assume there exists a constant $\alpha \leq \min_{g \in \mathcal{G}} \{\mathbb{E}_{\hat{\mu}} [e^{g(Z)}], \mathbb{E}_\mu [e^{g(Z)}]\}$. Then for $\forall \delta \in (0, 1)$ the following bound holds with probability at least $1 - \delta$,*

$$D_{KL}(\mu' || \mu) - D_{KL}(\hat{\mu}' || \hat{\mu}) \leq C_1(\kappa) \sqrt{\frac{1}{n} \left( d_1 \log \frac{2en}{d_1} + \log \frac{4}{\delta} \right)} + C_2(\alpha) \sqrt{\frac{1}{m} \left( d_2 \log \frac{2em}{d_2} + \log \frac{4}{\delta} \right)},$$

*where $C_1(\kappa) = \left(\frac{1}{2}\right)^{\frac{2-\kappa}{\kappa}} \left(\frac{\kappa}{\kappa-2}\right)^{\frac{\kappa-1}{\kappa}} \sup_{g \in \mathcal{G}} \sqrt[\kappa]{\mathbb{E}_\mu [g(Z)^\kappa]}$ and $C_2(\alpha) = \frac{2}{\alpha} \sup_{g \in \mathcal{G}} \mathbb{E}_\mu [e^{g(Z)}]$.*

*Proof.* Recall Lemma A.1, we have

$$D_{KL}(\mu'||\mu) = \sup_{g \in \mathcal{G}} \mathbb{E}_{\mu'}[g(Z')] - \log \mathbb{E}_\mu \left[ e^{g(Z)} \right],$$

and

$$D_{KL}(\hat\mu'||\hat\mu) = \sup_{g \in \mathcal{G}} \mathbb{E}_{\hat\mu'}[g(Z')] - \log \mathbb{E}_{\hat\mu} \left[ e^{g(Z)} \right].$$

Then, with the probability at least $1 - \delta$,

$$D_{KL}(\mu'||\mu) - D_{KL}(\hat\mu'||\hat\mu)$$

$$= \sup_{g \in \mathcal{G}} \mathbb{E}_{\mu'}[g(Z')] - \log \mathbb{E}_\mu \left[ e^{g(Z)} \right] - \left( \sup_{g \in \mathcal{G}} \mathbb{E}_{\hat\mu'}[g(Z')] - \log \mathbb{E}_{\hat\mu} \left[ e^{g(Z)} \right] \right)$$

$$\leq \sup_{g \in \mathcal{G}} \mathbb{E}_{\mu'}[g(Z')] - \log \mathbb{E}_\mu \left[ e^{g(Z)} \right] - \left( \mathbb{E}_{\hat\mu'}[g(Z')] - \log \mathbb{E}_{\hat\mu} \left[ e^{g(Z)} \right] \right)$$

$$= \sup_{g \in \mathcal{G}} \mathbb{E}_{\mu'}[g(Z')] - \mathbb{E}_{\hat\mu'}[g(Z')] + \log \mathbb{E}_{\hat\mu} \left[ e^{g(Z)} \right] - \log \mathbb{E}_\mu \left[ e^{g(Z)} \right]$$

$$\leq \sup_{g \in \mathcal{G}} |\mathbb{E}_{\mu'}[g(Z')] - \mathbb{E}_{\hat\mu'}[g(Z')]| + \sup_{g \in \mathcal{G}} \left| \log \mathbb{E}_{\hat\mu} \left[ e^{g(Z)} \right] - \log \mathbb{E}_\mu \left[ e^{g(Z)} \right] \right|$$

$$\leq \sup_{g \in \mathcal{G}} |\mathbb{E}_{\mu'}[g(Z')] - \mathbb{E}_{\hat\mu'}[g(Z')]| + \sup_{g \in \mathcal{G}} \frac{1}{\alpha} \left| \mathbb{E}_{\hat\mu} \left[ e^{g(Z)} \right] - \mathbb{E}_\mu \left[ e^{g(Z)} \right] \right| \tag{12}$$

$$\leq C_1(\kappa) \sqrt{\frac{1}{n} \left( d_1 \log \frac{2en}{d_1} + \log \frac{4}{\delta} \right)} + C_2(\alpha) \sqrt{\frac{1}{m} \left( d_2 \log \frac{2em}{d_2} + \log \frac{4}{\delta} \right)}, \tag{13}$$

where Eq. (12) is derived below.

W.L.O.G. assume that $\mathbb{E}_{\hat\mu} \left[ e^{g(Z)} \right] \leq \mathbb{E}_\mu \left[ e^{g(Z)} \right]$ (and Eq. (12) still holds when $\mathbb{E}_{\hat\mu} \left[ e^{g(Z)} \right] \geq \mathbb{E}_\mu \left[ e^{g(Z)} \right]$), then

$$\left| \log \mathbb{E}_{\hat\mu} \left[ e^{g(Z)} \right] - \log \mathbb{E}_\mu \left[ e^{g(Z)} \right] \right| = \left| \log \frac{\mathbb{E}_\mu \left[ e^{g(Z)} \right]}{\mathbb{E}_{\hat\mu} \left[ e^{g(Z)} \right]} \right| = \left| \log \left( 1 + \frac{\mathbb{E}_\mu \left[ e^{g(Z)} \right]}{\mathbb{E}_{\hat\mu} \left[ e^{g(Z)} \right]} - 1 \right) \right|$$

$$\leq \left| \frac{\mathbb{E}_\mu \left[ e^{g(Z)} \right]}{\mathbb{E}_{\hat\mu} \left[ e^{g(Z)} \right]} - 1 \right|$$

$$= \left| \frac{1}{\mathbb{E}_{\hat\mu} \left[ e^{g(Z)} \right]} \left( \mathbb{E}_\mu \left[ e^{g(Z)} \right] - \mathbb{E}_{\hat\mu} \left[ e^{g(Z)} \right] \right) \right|$$

$$\leq \frac{1}{\alpha} \left| \mathbb{E}_\mu \left[ e^{g(Z)} \right] - \mathbb{E}_{\hat\mu} \left[ e^{g(Z)} \right] \right|.$$

Eq. (13) is by Lemma B.1 and Lemma B.2. This concludes the proof. $\square$

With Theorem B.1 and Theorem 4.1, we immediately have the following corollary.

**Corollary B.1.** *Let the conditions in Theorem B.1 and Theorem 4.1 hold, then for any $w \in \mathcal{W}$,*

$$\left| \widetilde{\mathrm{Err}}(w) \right| \leq \sqrt{2} R \sqrt{D_{KL}(\hat\mu'||\hat\mu) + C_1(\kappa) \sqrt{\frac{1}{n} \left( d_1 \log \frac{2en}{d_1} + \log \frac{4}{\delta} \right)} + C_2(\alpha) \sqrt{\frac{1}{m} \left( d_2 \log \frac{2em}{d_2} + \log \frac{4}{\delta} \right)}},$$

*where $C_1(\kappa)$ and $C_2(\alpha)$ are the same as in Theorem B.1.*

## B.9 Additional Discussions on the Convergence of Empirical KL Divergence

Characterizing the convergence of the empirical KL divergence to the real KL is a challenging task that often requires several additional assumptions, as demonstrated in Theorem B.1. However, it is worth noting that the convergence rate of the empirical distribution to the real distribution in the KL sense is already established in the discrete space. This fact is supported by a classic result in (Cover & Thomas, 2006, Theorem 11.2.1), which we state in the following theorem:

**Theorem B.2.** *Let $\hat{\mu}$ and $\hat{\mu}'$ be defined as in Theorem B.1. Assume the space of $\mathcal{Z}$ is finite (i.e. $|\mathcal{Z}| \leq \infty$), then for $\forall \delta \in (0, 1)$, with probability at least $1 - \delta$,*

$$\mathrm{D}_{\mathrm{KL}}(\hat{\mu}||\mu) \leq \frac{|\mathcal{Z}|}{n} \log{(n+1)} + \frac{1}{n} \log{\frac{1}{\delta}}, \qquad \mathrm{D}_{\mathrm{KL}}(\hat{\mu}'||\mu') \leq \frac{|\mathcal{Z}|}{m} \log{(m+1)} + \frac{1}{m} \log{\frac{1}{\delta}}.$$

Thus, it suffices to ensure that the empirical KL converge to the real KL with the similar rate, although we do not know if there might exist a faster convergence rate.

### B.10 GENERALIZE TO APPROXIMATE TRIANGLE INEQUALITY

In Section 4, some results require that the loss obeys the triangle inequality (i.e. Assumption 4), such as Theorem 4.2 and Theorem 4.5. While the $0 - 1$ loss satisfies Assumption 4, some other loss may not. Thus, to generalize Theorem 4.2 and Theorem 4.5, we invoke an approximate triangle inequality, which is originally defined in Crammer et al. (2008).

**Assumption 5** ($\alpha$-Triangle). *$\ell(\cdot, \cdot)$ is symmetric and satisfies the following $\alpha$-triangle inequality: $\ell(y_1, y_2) \leq \alpha \left( \ell(y_1, y_3) + \ell(y_3, y_2) \right)$ for any $y_1, y_2, y_3 \in \mathcal{Y}$, where $\alpha \geq 1$ is a constant that may depend on the hypothesis space $\mathcal{W}$ and the loss $\ell$.*

**Remark B.1.** *We note that the squared loss satisfies 2-triangle inequality.*

Thus, Theorem 4.2 can be easily generalized below.

**Theorem B.3.** *If Assumption 5 holds and let $\ell(f_{w'}(X), f_w(X))$ be $R$-subgaussian for any $w, w' \in \mathcal{W}$. Then for any $w$,*

$$\widetilde{\mathrm{Err}}(w) \leq (\alpha^2 - 1)R_\mu + \alpha\sqrt{2R^2 \mathrm{D}_{\mathrm{KL}}(P_{X'}||P_X)} + \alpha^2 \lambda^*,$$

*where $\lambda^* = \min_{w \in \mathcal{W}} R_{\mu'}(w) + R_\mu(w)$.*

Theorem 4.5 can be generalized in the similar way. While Theorem B.3 strictly speaking is not a generalization bound, as it includes $R_\mu$ in the bound, it shares the same underlying concept as Theorem 4.2. Namely, to minimize the population risk in the target domain, it is essential for the source domain and target domain to be similar, and for both $R_\mu$ and $\lambda^*$ to be kept small.

## C OMITTED PROOFS AND ADDITIONAL DISCUSSIONS IN SECTION 5

### C.1 ADDITIONAL DISCUSSION ON THEOREM 5.1

To derive the bound in Theorem 5.1, we need to make use of the second equality in Eq. (1). In fact, by the definition of Err (the first equality in Eq. (1)), the unlabelled sample $S'_{X'_j}$ does not explicitly appear, so one can easily apply the similar information-theoretic analysis starting from the first equality in Eq. (1), and obtain an upper bound that consists of $I(W; Z_i)$ and $\mathrm{D}_{\mathrm{KL}}(\mu||\mu')$. Precisely, the following bound holds,

**Theorem C.1.** *Assume $\ell(f_w(X'), Y')$ is $R$-subgaussian for any $w \in \mathcal{W}$. Then*

$$|\mathrm{Err}| \leq \frac{1}{n} \sum_{i=1}^{n} \mathbb{E}\sqrt{2R^2 I(W; Z_i)} + \sqrt{2R^2 \mathrm{D}_{\mathrm{KL}}(\mu||\mu')}.$$

The proof of Theorem C.1 is nearly the same to the proof of (Wu et al., 2020, Corollary 2) and (Masiha et al., 2021, Corollary 1).

It's important to note that although

$$I(W; Z_i) \leq I(W; Z_i | X'_j) = \mathbb{E}_{X'_j}\left[ I^{X'_j}(W; Z_i) \right],$$

the bound in Theorem 5.1 is incomparable to the bound based on $I(W; Z_i)$. This is mainly due to the fact that we use the disintegrated version of mutual information, $I^{X'_j}(W; Z_i)$, and the expectation over $X'_j$ is outside of the square root, which is a convex function. Using $I^{X'_j}(W; Z_i)$ instead of

$I(W; Z_i)$ allows us to figure out more details about the role of unlabelled target data in the algorithm. Additionally, one can also prove a bound based on $I(W; Z_i | X'_j)$ (e.g., simply applying Jensen's inequality to Theorem 5.1), which is close to an individual and UDA version of (Bu et al., 2022, Theorem 3).

Furthermore, the first term in Theorem 5.1 characterize the expected generalization gap on the source domain (i.e. $\mathbb{E}_{W,S}[R_\mu(W) - R_S(W)]$), then the bound suggests us that it's possible to invoke the unlabelled target data to further improve the performance on source domain, and the simplest case is the semi-supervised learning (when $\mu = \mu'$).

**Compared with (Wu et al., 2020; Jose & Simeone, 2021b).** Notably, bounds in (Wu et al., 2020; Jose & Simeone, 2021b) fail to characterize the dependence between $W$ and $S'_{X'}$. More precisely, the algorithm-dependent term in their bounds is $I(W; Z_i)$ or $I(W; S)$, while our algorithm-dependent term is $I^{X'_j}(W; Z_i)$ that directly depends on the unlabelled target data. Moreover, while the disintegrated mutual information $I^{X'_j}(W; Z_i)$ and the unconditional mutual information $I(W; Z_i)$ cannot be directly compared, recent work by Wang & Mao (2023) provides empirical evidence comparing similar terms in the supervised learning setting. Specifically, they demonstrate that when the empirical risk is small, such as in a realizable case, the disintegrated mutual information is smaller than the unconditional mutual information. Conversely, when the empirical risk is large, the unconditional mutual information is the smaller of the two.

**More Discussion on the Vanishing of $I(X'_j; Z_i | W)$ in Remark 5.1.** Note that $S$ depends on $S'_{X'}$ given $W$, so intuitively the dependence between each individual instance $Z_i$ and $X'_j$ is weaker when $n$ and $m$ become larger. More precisely, W.L.O.G let $i = j = 1$, and recall that $W = \mathcal{A}(S'_{X'}, S)$, when $n, m \to \infty$, taking $S$ and $S'_{X'}$ as the input of the algorithm is nearly equivalent to computing $W$ based on the source distribution $\mu$ and the target distribution $P_{X'}$, thus, $W$ will only depend on the two distributions, without depending on the realizations $Z_1$ and $X'_1$ drawn respectively from the two distributions, that is, $I(Z_1; X'_1 | W) = I(Z_1; X'_1 | \mathcal{A}(\mu, P_{X'})) = I(Z_1; X'_1) = 0$. In addition, one may argue that what if $W = constant$ that does not really depend on the input data. In this case, $I(Z_1; X'_1 | W) = I(Z_1; X'_1) = 0$ will hold trivially. In the other extreme, if $n = 1$ and $m = 1$, then $W = \mathcal{A}(X'_1, Z_1)$, and the quantity $I(Z_1; X'_1 | \mathcal{A}(X'_1, Z_1))$ should be large. When $n$ and $m$ increase, it becomes $I(Z_1; X'_1 | \mathcal{A}(X'_{1:m}, Z_{1:n}))$. Now we want to guess $Z_1$ from $X'_1$, this should be easier when having the knowledge of $\mathcal{A}(X'_1; Z_1)$ compared with when having the knowledge of $\mathcal{A}(X'_{1:m}; Z_{1:n})$.

## C.2 PROOF OF THEOREM 5.1

*Proof.* By Lemma A.1,

$$
\begin{aligned}
&\mathrm{D}_{\mathrm{KL}}\left(P_{W,Z_i|X'_j=x'_j} || P_{W,Z'|X'_j=x'_j}\right) \\
=&\mathrm{D}_{\mathrm{KL}}\left(P_{W,Z_i|X'_j=x'_j} || P_{W|X'_j=x'_j} P_{Z'}\right) \\
\geq& \sup_t \mathbb{E}_{P_{W,Z_i|X'_j=x'_j}}[t\ell(f_W(X_i), Y_i)] - \log \mathbb{E}_{P_{W|X'_j=x'_j} P_{Z'}}[\exp(t\ell(f_W(X'), Y'))] \\
\geq& \sup_t \mathbb{E}_{P_{W,Z_i|X'_j=x'_j}}[t\ell(f_W(X_i), Y_i)] - \mathbb{E}_{P_{W|X'_j=x'_j}}[tR_{\mu'}(W)] - R^2 t^2/2,
\end{aligned}
\tag{14}
$$

where Eq. (14) is by the independence between algorithm output $W$ and unseen target domain data $Z'$, and the last inequality is by the subgaussian assumption.

Thus,

$$
\left| \mathbb{E}_{P_{W,Z_i|X'_j=x'_j}}[\ell(f_W(X_i), Y_i)] - \mathbb{E}_{P_{W|X'_j=x'_j}}[R_{\mu'}(W)] \right| \leq \sqrt{2R^2 \mathrm{D}_{\mathrm{KL}}\left(P_{W,Z_i|X'_j=x'_j} || P_{W|X'_j=x'_j} P_{Z'}\right)}.
\tag{15}
$$

Exploiting the fact that

$$
\begin{aligned}
|\text{Err}| &= \left| \frac{1}{n} \sum_{i=1}^{n} \mathbb{E}_{W,Z_i} \left[ \ell(f_W(X_i), Y_i) \right] - \mathbb{E}_{W,Z'} \left[ \ell(f_W(X'), Y') \right] \right| \\
&= \left| \frac{1}{m} \sum_{j=1}^{m} \mathbb{E}_{X'_j} \left[ \frac{1}{n} \sum_{i=1}^{n} \mathbb{E}_{W,Z_i|X'_j} \left[ \ell(f_W(X_i), Y_i) \right] - \mathbb{E}_{W,Z'|X'_j} \left[ \ell(f_W(X'), Y') \right] \right] \right| \\
&\leq \frac{1}{m} \sum_{j=1}^{m} \mathbb{E}_{X'_j} \left| \frac{1}{n} \sum_{i=1}^{n} \mathbb{E}_{W,Z_i|X'_j} \left[ \ell(f_W(X_i), Y_i) \right] - \mathbb{E}_{W,Z'|X'_j} \left[ \ell(f_W(X'), Y') \right] \right| \\
&\leq \frac{1}{nm} \sum_{j=1}^{m} \sum_{i=1}^{n} \mathbb{E}_{X'_j} \left| \mathbb{E}_{W,Z_i|X'_j} \left[ \ell(f_W(X_i), Y_i) \right] - \mathbb{E}_{W|X'_j} \left[ R_{\mu'}(W) \right] \right|,
\end{aligned}
$$

where the last two inequalities are by the Jensen's inequality for the absolute function.

Notice that

$$
\begin{aligned}
\mathrm{D_{KL}} \left( P_{W,Z_i|X'_j = x'_j} || P_{W|X'_j = x'_j} P_{Z'} \right) &= \mathbb{E}_{P_{W,Z_i|X'_j = x'_j}} \left[ \log \frac{P_{W,Z_i|X'_j = x'_j}}{P_{W|X'_j = x'_j} P_{Z'}} \right] \\
&= \mathbb{E}_{P_{W,Z_i|X'_j = x'_j}} \left[ \log \frac{P_{W|Z_i, X'_j = x'_j} P_{Z_i}}{P_{W|X'_j = x'_j} P_{Z'}} \right] \\
&= \mathbb{E}_{P_{W,Z_i|X'_j = x'_j}} \left[ \log \frac{P_{W|Z_i, X'_j = x'_j}}{P_{W|X'_j = x'_j}} \right] + \mathbb{E}_{P_{Z_i}} \left[ \log \frac{P_{Z_i}}{P_{Z'}} \right] \\
&= I(W; Z_i | X'_j = x'_j) + \mathrm{D_{KL}}(\mu || \mu').
\end{aligned}
$$

Recall Eq. (15), we then have

$$
\begin{aligned}
|\text{Err}| &\leq \frac{1}{nm} \sum_{j=1}^{m} \sum_{i=1}^{n} \mathbb{E}_{X'_j} \left| \mathbb{E}_{W,Z_i|X'_j} \left[ \ell(f_W(X_i), Y_i) \right] - \mathbb{E}_{W|X'_j} \left[ R_{\mu'}(W) \right] \right| \\
&\leq \frac{1}{nm} \sum_{j=1}^{m} \sum_{i=1}^{n} \mathbb{E}_{X'_j} \sqrt{2R^2 \mathrm{D_{KL}} \left( P_{W,Z_i|X'_j} || P_{W|X'_j} P_{Z'} \right)} \\
&= \frac{1}{nm} \sum_{j=1}^{m} \sum_{i=1}^{n} \mathbb{E}_{X'_j} \sqrt{2R^2 (I^{X'_j}(W; Z_i) + \mathrm{D_{KL}}(\mu || \mu'))} \\
&\leq \frac{1}{nm} \sum_{j=1}^{m} \sum_{i=1}^{n} \mathbb{E}_{X'_j} \sqrt{2R^2 I^{X'_j}(W; Z_i)} + \sqrt{2R^2 \mathrm{D_{KL}}(\mu || \mu')}.
\end{aligned}
$$

This completes the proof. $\qquad\square$

## C.3   PROOF OF COROLLARY 5.1

*Proof.* We now modify the proof in Theorem 5.1.

Recall that

$$
|\text{Err}| \leq \frac{1}{nm} \sum_{j=1}^{m} \sum_{i=1}^{n} \mathbb{E}_{X'_j} \left| \mathbb{E}_{W,Z_i|X'_j} \left[ \ell(f_W(X_i), Y_i) \right] - \mathbb{E}_{W|X'_j} \left[ R_{\mu'}(W) \right] \right|.
$$

We first decompose the right hand side,

$$
\left| \mathbb{E}_{W,Z_i|X'_j=x'_j} \left[ \ell(f_W(X_i),Y_i) \right] - \mathbb{E}_{W|X'_j=x'_j} \left[ R_{\mu'}(W) \right] \right|
$$
$$
= \left| \mathbb{E}_{W,Z_i|X'_j=x'_j} \left[ \ell(f_W(X_i),Y_i) \right] - \mathbb{E}_{W|X'_j=x'_j} \left[ R_\mu(W) \right] + \mathbb{E}_{W|X'_j=x'_j} \left[ R_\mu(W) \right] - \mathbb{E}_{W|X'_j=x'_j} \left[ R_{\mu'}(W) \right] \right|
$$
$$
\leq \left| \mathbb{E}_{W,Z_i|X'_j=x'_j} \left[ \ell(f_W(X_i),Y_i) \right] - \mathbb{E}_{W|X'_j=x'_j} \left[ R_\mu(W) \right] \right| + \left| \mathbb{E}_{W|X'_j=x'_j} \left[ R_\mu(W) - R_{\mu'}(W) \right] \right|
$$
$$
\leq \left| \mathbb{E}_{W,Z_i|X'_j=x'_j} \left[ \ell(f_W(X_i),Y_i) \right] - \mathbb{E}_{W|X'_j=x'_j} \left[ R_\mu(W) \right] \right| + \frac{M}{\sqrt{2}} \sqrt{\min\{\mathrm{D_{KL}}(\mu||\mu'), \mathrm{D_{KL}}(\mu'||\mu)\}},
$$

where the last inequality is by Corollary 4.2.

Then for the first term in RHS, notice that

$$
\mathrm{D_{KL}}\left( P_{W,Z|X'_j=x'_j} || P_{W,Z_i|X'_j=x'_j} \right)
$$
$$
= \mathrm{D_{KL}}\left( P_{W|X'_j=x'_j} P_Z || P_{W,Z_i|X'_j=x'_j} \right)
$$
$$
\geq \sup_t \mathbb{E}_{P_{W|X'_j=x'_j} P_Z} \left[ t\ell(f_W(X),Y) \right] - \log \mathbb{E}_{P_{W,Z_i|X'_j=x'_j}} \left[ \exp t\ell(f_W(X_i),Y_i) \right]
$$
$$
\geq \sup_t \mathbb{E}_{P_{W|X'_j=x'_j} P_Z} \left[ t\ell(f_W(X),Y) \right] - \mathbb{E}_{P_{W,Z_i|X'_j=x'_j}} \left[ t\ell(f_W(X_i),Y_i) \right]
$$
$$
- \log \mathbb{E}_{P_{W,Z_i|X'_j=x'_j}} \left[ e^{t(\ell(f_W(X_i),Y_i) - \mathbb{E}_{P_{W,Z_i|X'_j=x'_j}} [\ell(f_W(X_i),Y_i)])} \right]
$$
$$
\geq \sup_t \mathbb{E}_{P_{W|X'_j=x'_j}} \left[ t R_\mu(W) \right] - \mathbb{E}_{P_{W,Z_i|X'_j=x'_j}} \left[ t\ell(f_W(X_i),Y_i) \right] - M^2 t^2/8,
$$

where the last inequality is due to the fact that $\ell$ is bounded by $M$ and $\ell(f_W(X_i),Y_i)$ is $M/2$-subgaussian.

Thus,

$$
\left| \mathbb{E}_{W,Z_i|X'_j=x'_j} \left[ \ell(f_W(X_i),Y_i) \right] - \mathbb{E}_{W|X'_j=x'_j} \left[ R_\mu(W) \right] \right| \leq \sqrt{\frac{M^2}{2} \mathrm{D_{KL}}\left( P_{W|X'_j=x'_j} P_Z || P_{W,Z_i|X'_j=x'_j} \right)}
$$
$$
= \sqrt{\frac{M^2}{2} L\left( W, Z_i | X'_j = x'_j \right)}.
$$

Plugging this inequality with the decomposition into the inequality at the beginning of the proof, we have

$$
|\mathrm{Err}| \leq \frac{1}{nm} \sum_{j=1}^m \sum_{i=1}^n \mathbb{E}_{X'_j} \sqrt{\frac{M^2}{2} L^{X'_j}(W, Z_i)} + \frac{M}{\sqrt{2}} \sqrt{\min\{\mathrm{D_{KL}}(\mu||\mu'), \mathrm{D_{KL}}(\mu'||\mu)\}}.
$$

Similar development also holds for $\mathrm{D_{KL}}\left( P_{W,Z_i|X'_j=x'_j} || P_{W|X'_j=x'_j} P_Z \right)$ as in the proof of Theorem 5.1, thus

$$
|\mathrm{Err}| \leq \frac{M}{\sqrt{2}nm} \sum_{j=1}^m \sum_{i=1}^n \mathbb{E}_{X'_j} \sqrt{\min\left\{ I^{X'_j}(W;Z_i), L^{X'_j}(W;Z_i) \right\}} + \frac{M}{\sqrt{2}} \sqrt{\min\left\{ \mathrm{D_{KL}}(\mu||\mu'), \mathrm{D_{KL}}(\mu'||\mu) \right\}}.
$$

This completes the proof. $\qquad\square$

## C.4 PROOF OF THEOREM 5.2

*Proof.* Similar to the proof of Corollary 5.1, recall Theorem 4.4,

$$|\text{Err}|$$

$$\leq \frac{1}{nm} \sum_{j=1}^{m} \sum_{i=1}^{n} \mathbb{E}_{X_j'} \left| \mathbb{E}_{W,Z_i|X_j'} \left[ \ell(f_W(X_i), Y_i) \right] - \mathbb{E}_{W|X_j'} \left[ R_\mu(W) \right] + \mathbb{E}_{W|X_j'} \left[ R_\mu(W) \right] - \mathbb{E}_{W|X_j'} \left[ R_{\mu'}(W) \right] \right|$$

$$\leq \frac{1}{nm} \sum_{j=1}^{m} \sum_{i=1}^{n} \mathbb{E}_{X_j'} \left| \mathbb{E}_{W,Z_i|X_j'} \left[ \ell(f_W(X_i), Y_i) \right] - \mathbb{E}_{W|X_j'} \left[ R_\mu(W) \right] \right| + \beta \mathbb{W}(\mu, \mu')$$

$$\leq \frac{1}{nm} \sum_{j=1}^{m} \sum_{i=1}^{n} \mathbb{E}_{X_j', Z_i} \left| \mathbb{E}_{W|Z_i, X_j'} \left[ \ell(f_W(X_i), Y_i) \right] - \mathbb{E}_{W|X_j'} \left[ \ell(f_W(X_i), Y_i) \right] \right| + \beta \mathbb{W}(\mu, \mu')$$

$$\leq \frac{\beta'}{nm} \sum_{j=1}^{m} \sum_{i=1}^{n} \mathbb{E}_{X_j', Z_i} \mathbb{W}(P_{W|X_j', Z_i}, P_{W|X_j'}) + \beta \mathbb{W}(\mu, \mu'),$$

where the last inequality is by Lemma A.3. This concludes the proof. □

## C.5 PROOF OF COROLLARY 5.2

*Proof.* Similar to the proof of Corollary 4.3, replacing Wasserstein distance by the total variation and replacing $\beta$ and $\beta'$ by $M$, will give us the first inequality,

$$\left| \widetilde{\text{Err}} \right| \leq \frac{M}{nm} \sum_{j=1}^{m} \sum_{i=1}^{n} \mathbb{E}_{X_j', Z_i} \left[ \text{TV}(P_{W|Z_i, X_j'}, P_{W|X_j'}) \right] + M \text{TV}(\mu, \mu').$$

The second inequality is by Lemma A.4,

$$\left| \widetilde{\text{Err}} \right| \leq \frac{M}{nm} \sum_{j=1}^{m} \sum_{i=1}^{n} \mathbb{E}_{X_j', Z_i} \sqrt{\frac{1}{2} \text{D}_{\text{KL}}(P_{W|Z_i, X_j'} || P_{W|X_j'})} + \sqrt{\frac{M^2}{2} \text{D}_{\text{KL}}(\mu || \mu')}.$$

Again, one can also apply Lemma A.5 here. This concludes the proof. □

## C.6 PROOF OF THEOREM 5.3

*Proof.* Recall Theorem 5.1 and by Jensen's inequality we have

$$|\text{Err}| \leq \frac{1}{nm} \sum_{j=1}^{m} \sum_{i=1}^{n} \mathbb{E}_{X_j'} \sqrt{2R^2 I^{X_j'}(W; Z_i)} + \sqrt{2R^2 \text{D}_{\text{KL}}(\mu || \mu')}$$

$$\leq \sqrt{\frac{2R^2}{nm} \sum_{j=1}^{m} \sum_{i=1}^{n} I(W; Z_i | X_j')} + \sqrt{2R^2 \text{D}_{\text{KL}}(\mu || \mu')}.$$

Let $X_{1,\ldots,j-1,j+1,\ldots,m}' = S_{X'}' \setminus X_j'$. Notice that

$$I(W; Z_i | S_{X'}') = I(W; Z_i | S_{X'}') + I(X_{1,\ldots,j-1,j+1,\ldots,m}'; Z_i | X_j')$$
$$= I(W; Z_i | X_j') + I(X_{1,\ldots,j-1,j+1,\ldots,m}'; Z_i | X_j', W)$$
$$\geq I(W; Z_i | X_j').$$

Thus, $I(W; Z_i | X_j') \leq I(W; Z_i | S_{X'}')$. Then

$$\frac{1}{nm} \sum_{j=1}^{m} \sum_{i=1}^{n} I(W; Z_i | X_j') \leq \frac{1}{nm} \sum_{j=1}^{m} \sum_{i=1}^{n} I(W; Z_i | S_{X'}') = \frac{1}{n} \sum_{i=1}^{n} I(W; Z_i | S_{X'}').$$

Then, since $S \perp\!\!\!\perp S'_{X'}$ and $Z_i \perp\!\!\!\perp Z_{1:i-1}$ for any $i \in [n]$, by the chain rule of mutual information, we have

$$
\begin{aligned}
I(W; S|S'_{X'}) = \sum_{i=1}^{n} I(W; Z_i|S'_{X'}, Z_{1:i-1}) &= \sum_{i=1}^{n} I(W; Z_i|S'_{X'}, Z_{1:i-1}) + I(Z_i; Z_{1:i-1}) \\
&= \sum_{i=1}^{n} I(W, Z_{1:i-1}; Z_i|S'_{X'}) \\
&= \sum_{i=1}^{n} I(W; Z_i|S'_{X'}) + I(Z_i; Z_{1:i-1}|S'_{X'}, W) \\
&\geq \sum_{i=1}^{n} I(W; Z_i|S'_{X'}).
\end{aligned}
$$

Thus, the generalization error bound becomes

$$
|\mathrm{Err}| \leq \sqrt{\frac{2R^2}{n} I(W; S|S'_{X'})} + \sqrt{2R^2 \mathrm{D}_{\mathrm{KL}}(\mu||\mu')}.
$$

Recall the updating rule of $W$ and notice that $W_0$ is independent of $S$ and $S'_{X'}$, the following process is by using the chain rule of mutual information and data processing inequality recurrently,

$$
\begin{aligned}
I(W_T; S|S'_{X'}) &= I(W_{T-1} - \eta_T g(W_{T-1}, Z_{B_T}, X'_{B_T}) + N_T; S|S'_{X'}) \\
&\leq I(W_{T-1}, -\eta_T g(W_{T-1}, Z_{B_T}, X'_{B_T}) + N_T; S|S'_{X'}) \\
&= I(W_{T-1}; S|S'_{X'}) + I(\eta_T g(W_{T-1}, Z_{B_T}, X'_{B_T}) + N_T; S|S'_{X'}, W_{T-1}) \\
&\vdots \\
&= \sum_{t=1}^{T} I(\eta_t g(W_{t-1}, Z_{B_t}, X'_{B_t}) + N_t; S|S'_{X'}, W_{t-1}).
\end{aligned}
$$

For each $t \in [T]$, denote $g(W_{t-1}, Z_{B_t}, X'_{B_t})$ as $G_t$, then

$$
\begin{aligned}
I(\eta_t g(W_{t-1}, Z_{B_t}, X'_{B_t}) + N_t; S|S'_{X'}, W_{t-1}) &= \mathbb{E}_{S'_{X'}, W_{t-1}, S}\left[\mathrm{D}_{\mathrm{KL}}(P_{G_t + \frac{N_t}{\eta_t}|S, S'_{X'}, W_{t-1}}||P_{G_t + \frac{N_t}{\eta_t}|S'_{X'}, W_{t-1}})\right] \\
&\leq \mathbb{E}_{S'_{X'}, W_{t-1}, S}\left[\mathrm{D}_{\mathrm{KL}}(P_{G_t + \frac{N_t}{\eta_t}|S, S'_{X'}, W_{t-1}}||P_{\mathbb{E}_S[G_t] + \frac{N_t}{\eta_t}|S'_{X'}, W_{t-1}})\right] \\
&= \frac{\eta_t^2}{2\sigma_t^2} \mathbb{E}_{S'_{X'}, W_{t-1}, S}\left[||G_t - \mathbb{E}_S[G_t]||^2\right],
\end{aligned}
$$

where the inequality is by Lemma A.6 and the last equality is by the KL divergence between two Gaussian distributions.

Finally, putting everything together,

$$
|\mathrm{Err}| \leq \sqrt{\frac{R^2}{n} \sum_{t=1}^{T} \frac{\eta_t^2}{\sigma_t^2} \mathbb{E}_{S'_{X'}, W_{t-1}, S}\left[||G_t - \mathbb{E}_S[G_t]||^2\right]} + \sqrt{2R^2 \mathrm{D}_{\mathrm{KL}}(\mu||\mu')},
$$

which concludes the proof. $\qquad\square$

## C.7   DERIVATION OF EQ. (6)

Recall the expected cross-entropy loss, we have

$$
\begin{aligned}
\mathbb{E}_{W,Z_i}\left[\ell(f_W(T_i), Y_i)\right] &= \mathbb{E}_{Z_i,W}\left[-\log Q_{Y_i|T_i,W}\right]\\
&= \mathbb{E}_{Z_i,W}\left[\log \frac{P_{Y_i|T_i,W}}{Q_{Y_i|T_i,W} P_{Y_i|T_i,W}}\right]\\
&= H(Y_i|T_i,W) + \mathbb{E}_{X_i,W}\left[\mathrm{D_{KL}}(P_{Y_i|T_i,W}||Q_{Y_i|T_i,W})\right]\\
&= \mathbb{E}_{Z_i,W}\left[\log \frac{P_{Y_i|T_i} P_{W|T_i}}{P_{Y_i|T_i,W} P_{Y_i|T_i} P_{W|T_i}}\right] + \mathbb{E}_{T_i,W}\left[\mathrm{D_{KL}}(P_{Y_i|T_i,W}||Q_{Y_i|T_i,W})\right]\\
&= \mathbb{E}_{Z_i,W}\left[\log \frac{P_{Y_i|T_i} P_{W|T_i}}{P_{Y_i,W|T_i} P_{Y_i|T_i}}\right] + \mathbb{E}_{T_i,W}\left[\mathrm{D_{KL}}(P_{Y_i|T_i,W}||Q_{Y_i|T_i,W})\right]\\
&= H(Y_i|T_i) - I(W;Y_i|T_i) + \mathbb{E}_{T_i,W}\left[\mathrm{D_{KL}}(P_{Y_i|T_i,W}||Q_{Y_i|T_i,W})\right]
\end{aligned}
$$

## C.8   ADDITIONAL DISCUSSION ON LIMIT

In Section 5, we discussed the LIMIT approach proposed by Harutyunyan et al. (2020) as a means of controlling label information memorization during training. Roughly speaking, to update the classifier parameters, LIMIT constructs an auxiliary network that predicts gradients instead of using the true gradients, which avoids direct use of the true labels for training. To obtain accurate gradients, the auxiliary network needs to be trained using the true labels. We found that the training of LIMIT is unstable and difficult to tune the hyperparameters when used under UDA settings. Therefore, we opted to use the pseudo label strategy proposed in Section 5 instead of the pseudo gradient strategy.

## D   EXPERIMENT DETAILS

We implemented our approach using PyTorch (Paszke et al., 2019) and conducted all experiments on NVIDIA Tesla V100 GPUs with 32 GB of memory. Our code builds largely on the implementation from Gulrajani & Lopez-Paz (2021)[3] and Nguyen et al. (2022)[4].

### D.1   OBJECTIVE FUNCTIONS OF GRADIENT PENALTY AND CONTROLLING LABEL INFORMATION

For every iteration, the objective function after adding the gradient penalty becomes

$$
\min_W \hat{L}(W, Z_{B_t}, X'_{B_t}) + \lambda_1 ||g(W, Z_{B_t}, X'_{B_t})||^2,
$$

where $\hat{L}(W, Z_{B_t}, X'_{B_t})$ is some loss function for the source and target domain data in the current mini-batch and $\lambda_1$ is the trade-off coefficient. For example, if we combine ERM with gradient penalty then $\hat{L}(W, Z_{B_t}, X'_{B_t}) = \frac{1}{|B_t|}\sum_{k\in B_t}\ell(f_W(X_k), Y_k)$ and $\ell$ could be the cross-entropy loss. Moreover, if we combine KL guided marginal alignment algorithm (Nguyen et al., 2022) with gradient penalty then the objective function is

$$
\min_{W,\theta} \frac{1}{|B_t|}\sum_{k\in B_t}\ell(f_W(T_k), Y_k) + \beta_1 \mathrm{D_{KL}}(P_{T'}||P_T) + \beta_2 \mathrm{D_{KL}}(P_T||P_{T'}) + \lambda_1 ||g(W, Z_{B_t}, X'_{B_t})||^2,
$$

where $\theta$ is the parameters of the representation network and the gradient is

$$
g(W, Z_{B_t}, X'_{B_t}) = \frac{1}{|B_t|}\sum_{k\in B_t}\nabla_{W,\theta}\ell(f_W(T_k), Y_k) + \beta_1 \nabla_\theta \mathrm{D_{KL}}(P_{T'}||P_T) + \beta_2 \nabla_\theta \mathrm{D_{KL}}(P_T||P_{T'}).
$$

In Nguyen et al. (2022), the representation distribution is modelled as an Gaussian distribution, i.e., $T \sim \mathcal{N}(\mu_\theta, \sigma_\theta^2 \mathrm{I}_d|X)$ and $T' \sim \mathcal{N}(\mu_\theta, \sigma_\theta^2 \mathrm{I}_d|X')$. Additionally, let the batch size be $b = |B_t|$, the

---

[3]Available at: https://github.com/facebookresearch/DomainBed.
[4]Available at: https://github.com/atuannguyen/kl.

Table 2: RotatedMNIST and Digits Experiments of **ERM-CL**. Results of ERM are reported from Nguyen et al. (2022).

| Method | RotatedMNIST ($0°$ as source domain) | | | | | | Digits | | | |
| | $15°$ | $30°$ | $45°$ | $60°$ | $75°$ | Ave | $M \rightarrow U$ | $U \rightarrow M$ | $S \rightarrow M$ | Ave |
|---|---|---|---|---|---|---|---|---|---|---|
| ERM | 97.5±0.2 | 84.1±0.8 | 53.9±0.7 | 34.2±0.4 | 22.3±0.5 | 58.4 | 73.1±4.2 | 54.8±6.2 | 65.9±1.4 | 64.6 |
| ERM-GP | **97.5±0.1** | **86.2±0.5** | **62.0±1.9** | **34.8±2.1** | **26.1±1.2** | **61.2** | **91.3±1.6** | **72.7±4.2** | 68.4±0.2 | 77.5 |
| ERM-CL | 97.3±0.1 | 84.1±0.1 | 56.9±2.5 | 34.2±1.9 | 25.5±1.6 | 59.6 | 88.9±0.4 | 71.2±3.6 | **73.5±1.4** | **77.9** |

empirical KL divergence is estimated by the mini-batch data, as given in Nguyen et al. (2022),

$$\beta_1 D_{KL}(P_{T'}||P_T) + \beta_2 D_{KL}(P_T||P_{T'})$$
$$\approx \beta_1 \frac{1}{b} \sum_{k \in B_t} [\log P_{T'_k} - \log P_{T_k}] + \beta_2 \frac{1}{b} \sum_{k \in B_t} [\log P_{T_k} - \log P_{T'_k}]$$
$$\approx \beta_1 \frac{1}{b} \sum_{k \in B_t} \left[ \log \frac{1}{b} \sum_{k \in B_t} P_{T'_k|X'_k} - \log \frac{1}{b} \sum_{k \in B_t} P_{T_k|X_k} \right] + \beta_2 \frac{1}{b} \sum_{k \in B_t} \left[ \log \frac{1}{b} \sum_{k \in B_t} P_{T_k|X_k} - \log \frac{1}{b} \sum_{k \in B_t} P_{T'_k|X'_k} \right],$$

where $P_{T_k|X_k} = \mathcal{N}(\mu_\theta, \sigma_\theta^2 I_d | X_k)$ and $P_{T'_k|X'_k} = \mathcal{N}(\mu_\theta, \sigma_\theta^2 I_d | X'_k)$. To be more precise, $\mu_\theta$ and $\sigma_\theta$ are the outputs of the representation network. Since the forward pass requires the sampling of $T_k$ and $T'_k$, we need to use the reparameterization trick (Kingma & Welling, 2013) for the backward pass.

When we train the model with controlling label information, the objective function becomes

$$\min_W \hat{L}(W, Z_{B_t}, X'_{B_t}) + \lambda_2 ||W - \widetilde{W}||^2,$$

where $\widetilde{W}$ is the auxiliary classifier and $\lambda_2$ is the trade-off hyperparameter.

Similarly, when we combine KL guided marginal alignment algorithm with controlling label information, then the objective function in every iteration is

$$\min_{W,\theta} \frac{1}{|B_t|} \sum_{k \in B_t} \ell(f_W(T_k), Y_k) + \beta_1 D_{KL}(P_{T'}||P_T) + \beta_2 D_{KL}(P_T||P_{T'}) + \lambda_2 ||W - \widetilde{W}||^2.$$

In addition, the training objective for the auxiliary classifier is

$$\min_{\widetilde{W}} \frac{1}{|B_t|} \sum_{k \in B_t} \ell(f_{\widetilde{W}}(T'_k), f_W(T'_k)) + \frac{1}{|B_t|} \sum_{k \in B_t} \ell(f_{\widetilde{W}}(T_k), f_W(T_k)). \tag{16}$$

In practice, removing the second term would not affect the performance. Note that we need to disenable the automatic differentiation of $T$, $T'$ and $W$ when executing the backward pass for the auxiliary classifier. The detailed algorithm of controlling label information is given in the next section.

### D.2 ALGORITHM OF CONTROLLING LABEL INFORMATION AND ADDITIONAL RESULTS OF ERM-CL

If we only provide the pseudo labels for the target domain data to the auxiliary classifier, i.e. removing the second term in Eq (16), the Algorithm 1 is the algorithm for combining any marginal alignment algorithm with controlling label information.

Even without incorporating with the marginal alignment algorithm, e.g., ERM, in which case $L_r$ is removed, Algorithm 1 still boosts the performance in practice.

Table 2 shows that **ERM-CL** can overall outperform the basic **ERM** and is close to the performance of **ERM-GP**.

---

**Algorithm 1** Controlling Label Information

---

**Require:** Source domain labelled dataset $S$, Target domain unlabelled dataset $S'_{X'}$, Batch size $b$, Classification loss function $\ell_c$, Marginal alignment loss function $\ell_r$, Initial classifier parameter $\boldsymbol{w}_0 = \widetilde{\boldsymbol{w}}_0$, Initial representation network parameter $\boldsymbol{\theta}_0$, Learning rate $\eta$, Lagrange multiplier $\lambda_2$

    **while** $\boldsymbol{w}_t, \theta_t$ not converged **do**

2:    Update iteration: $t \leftarrow t + 1$

      Sample $\mathcal{Z}_{\mathcal{B}} = \{\boldsymbol{z}_i\}_{i=1}^b$ from source domain training set $S$

4:    Sample $\mathcal{X}'_{\mathcal{B}} = \{\boldsymbol{x}'_i\}_{i=1}^b$ from target domain training set $S'_{X'}$

      Compute distance from the auxiliary classifier $\boldsymbol{dis} \leftarrow ||\boldsymbol{w}_t - \widetilde{\boldsymbol{w}}_t||^2$

6:    Compute marginal alignment loss $L_r \leftarrow \frac{1}{b} \sum_{i=1}^b \ell_r(\theta_t, \boldsymbol{z}_i, \boldsymbol{x}'_i)$

      Compute classification loss $L_c \leftarrow \frac{1}{b} \sum_{i=1}^b \ell_c(\boldsymbol{w}_t, \theta_t, \boldsymbol{z}_i, \boldsymbol{x}'_i)$

8:    Compute gradient:

      $g_{\mathcal{B}} \leftarrow \nabla(L_c + L_r + \lambda_2 \boldsymbol{dis})$

      Update parameter: $\boldsymbol{w}_{t+1} \leftarrow \boldsymbol{w}_t - \eta \cdot g_{\mathcal{B}}$, $\boldsymbol{\theta}_{t+1} \leftarrow \boldsymbol{\theta}_t - \eta \cdot g_{\mathcal{B}}$

10:   Obtain the pseudo labels $\mathcal{Y}'_{\mathcal{B}} \leftarrow f_{\boldsymbol{w}_t}(g_{\boldsymbol{\theta}_t}(\mathcal{X}'_{\mathcal{B}}))$

      Compute auxiliary classification loss $L_a \leftarrow \frac{1}{b} \sum_{i=1}^b \ell_c(\widetilde{\boldsymbol{w}}_t, \theta_t, \boldsymbol{x}'_i, \boldsymbol{y}'_i)$

12:   Compute auxiliary classifier gradient:

      $\widetilde{g}_{\mathcal{B}} \leftarrow \nabla L_a$

      Update auxiliary classifier parameter: $\widetilde{\boldsymbol{w}}_{t+1} \leftarrow \widetilde{\boldsymbol{w}}_t - \eta \cdot \widetilde{g}_{\mathcal{B}}$

14: **end while**

---

Table 3: Ablation study on Effect of Gradient Penalty Hyperparameter.

| $\lambda_1$ | 0 | 0.1 | 0.3 | 0.5 |
|---|---|---|---|---|
| $0° \rightarrow 60°$ | 75.5±2.4 | 88.0±8.1 | 82.8±5.8 | 80.1±3.7 |
| $\mathbf{S} \rightarrow \mathbf{M}$ | 92.5±0.9 | 93.6±1.2 | 93.8±1.1 | 93.1±1.7 |

### D.3 ARCHITECTURES AND HYPERPARAMETERS

The network architecture in this work is the same as in Gulrajani & Lopez-Paz (2021) and Nguyen et al. (2022), where a simple CNN is used.

Other settings are also the same as Gulrajani & Lopez-Paz (2021) and Nguyen et al. (2022), for example, each algorithm is trained for 100 epochs. To select the hyperparameters ($\lambda_1$ and $\lambda_2$) for **ERM-GP**, **ERM-KL**, **KL-GP** and **KL-CL**, we perform random search. Specifically, $\lambda_1$ is searched between $[0.1, 0.9]$ and $\lambda_2$ is searched between $[10^{-6}, 0.8]$. Other hyperparameters searching range could be found in the source code of Nguyen et al. (2022).

### D.4 ADDITIONAL EXPERIMENTAL RESULTS

### D.5 ABLATION STUDY ON THE EFFECT OF GRADIENT PENALTY HYPERPARAMETER

Our study includes an ablation analysis to investigate the impact of the hyperparameter $\lambda_1$ in the context of **KL-GP**. Specifically, we conduct experiments on both RotatedMNIST and Digits datasets, where the source and target domains are set to 0°/60° and SVHN/MNIST, respectively. Table 3 summarizes the results. It is worth noting that setting $\lambda_1$ to zero effectively reduces **KL-GP** to **KL**, and our results confirm the efficacy of including the gradient penalty term in **KL-GP**.

### D.6 VISUALIZATION RESULTS

To visualize the representations of models trained using **KL**, **KL-GP**, and **KL-CL**, we employ t-SNE (Van der Maaten & Hinton, 2008). Figure 2 displays the visualization results when SVHN is used as the source domain and MNIST as the target domain. Our findings indicate that the incorporation of additional regularizers yields a slight improvement in representation alignment. However, it

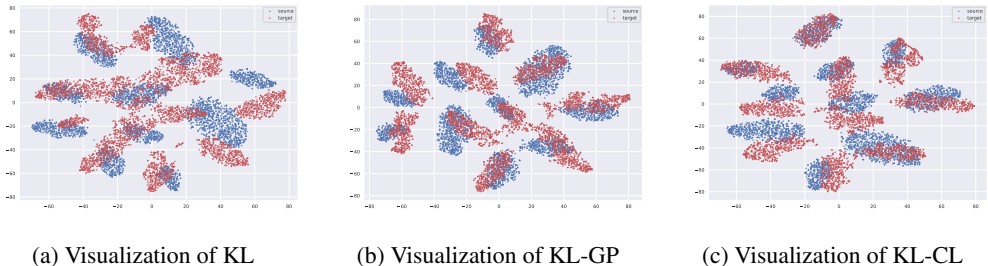

(a) Visualization of KL          (b) Visualization of KL-GP          (c) Visualization of KL-CL

Figure 2: Visualization results of representations obtained by using t-SNE. The source domain (blue points) is SVHN and the target domain (red points) is MNIST.

Table 4: VisDA17 experiments. Results of baselines are reported directly from Nguyen et al. (2022).

| Method | Synthetic → Real |
|--------|------------------|
| ERM | 39.1±0.5 |
| DANN | 57.7±1.3 |
| MMD | 62.8±1.1 |
| CORAL | 39.5±4.5 |
| WD | 38.9±4.8 |
| KL | 70.6±0.5 |
| KL-GP | **71.9±0.7** |
| KL-CL | 71.3±0.4 |

is essential to note that these regularization terms are primarily designed to enhance the performance of the classifier network, rather than the representation network.

### D.7 RESULTS ON VISDA17

We also conduct experiments on the VisDA17 dataset (Peng et al., 2017), which is a real-world classification task with $280K$ images from 12 classes. Particularly, the source domain contains synthetic images and the target domain contains real images. Table 4 presents our experimental results. Notably, our regularization techniques, namely **KL-GP** and **KL-CL** are still capable of improving the performance of the KL guided marginal alignment algorithm to some extent.

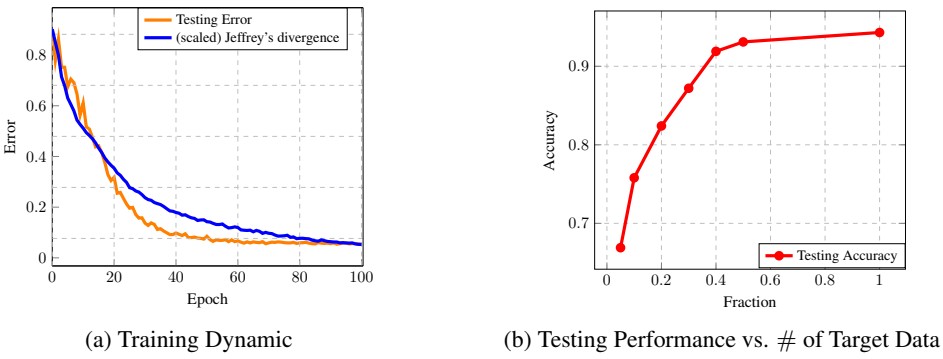

(a) Training Dynamic          (b) Testing Performance vs. # of Target Data

Figure 3: **KL** on **S→M**. The left figure is the comparison of the Jeffrey's divergence in the representation space and the testing error. The right figure is the evolution of testing accuracy with respect to the different fraction of unlabelled target data used for training.

## D.8 DYNAMICS OF JEFFREY'S DIVERGENCE

The representation space version of Corollary 4.2 suggests that a small Jeffrey's divergence can lead to a low testing error. Figure 3a demonstrates that the dynamic of Jeffrey's divergence, as computed in the representation space, can effectively characterize the evolution of the testing error throughout the training phase. Additionally, Figure 3b reveals that the number of target data used has an impact on testing performance. Specifically, when less than half of the available unlabelled target data is used, performance increases with the number of data. However, when more than half of the unlabelled target data is used, there is only marginal improvement on performance.

