# OpenReview forum: "Information-Theoretic Analysis of Unsupervised Domain Adaptation"
_ICLR.cc/2023/Conference — ICLR 2023 poster_

### Official Review · Reviewer_mqeg · 2022-10-18

**Confidence:** 4
**Correctness:** 3
**Technical Novelty And Significance:** 3
**Empirical Novelty And Significance:** 3
**Recommendation:** 6

**Clarity, Quality, Novelty And Reproducibility:**

* **Clarity**: The paper is clear and well-written.

* **Quality**: The quality of the paper is good. As mentioned in the review, the resulting bounds are tighter and some of them reveal interesting insights. Also, the regularization scheme empirically performs better than previous schemes in the literature.

* **Novelty**: While the techniques are not novel and have been employed previously (see weaknesses), their use in the UDA is novel as far as I know. Nonetheless, the resulting bounds and insights are indeed new, as well as the regularization technique.

* **Reproducibility**: \
    *Theory*: I reproduced all the proofs except for the questions that I placed the authors in the weaknesses.

    *Experiments*: There is code provided to reproduce the experiments.  I hope this code is then uploaded to some easily accessible repository and with an easy link in the main text.

**Strength And Weaknesses:**

**Strengths**

* The paper combines nicely the known techniques to bound the classical generalization error in the UDA setting. Typically these are only used to bound the PP error and not the EP error.

* The resulting bounds are often tighter than those in the literature and some of them provide new insights into the UDA problem.

* The resulting regularization strategies obtain better results than the known schemes to improve domain adaptation.

**Weaknesses**

* Although the classical generalization error papers introducing the techniques used to bound the difference of expectations are quickly cited in the related work section, it is not clear in the main text where and how these techniques are used or how they inspired the bounds in the main text.

    * Section 4.1. is strongly powered by Lemma A.1. which is basically Lemma 1 in *[Xu and Raginsky 2017]* letting $\mathcal{X} \times \mathcal{Y} = \Theta$, $P_{X,Y} = Q$, and $P_{X} \otimes P_{Y} = P$.
    * Section 4.2. (Theorem 4.4., Corollary 4.3, and the discussion around them) is based on *[Rodríguez-Gálvez et al. 2021]*.
    * Section 5.1. is first based on the disintegration ideas from *[Negrea et al. 2019]* and then on the decompositions of *[Rodríguez-Gálvez et al. 2021]*.
    * Section 5.2. is based on the information-theoretic analyses of SGLD and noisy SGD *[Pensia et al. 2018, Bu et al. 2019, Negrea et al. 2019, Haghifam et al. 2020, Rodríguez-Gálvez et al. 2021b, Neu et al. 2021, Wang and Mao 2022]*.
    * Also in Section 5.2., it is stated that gradient penalty is strongly theoretically justified for standard supervised learning based on Theorem 5.3. Similar observations can be made from the results on standard generalization error from both the information-theoretic *[Negrea et al. 2019, Haghifam et al. 2020, Rodríguez-Gálvez et al. 2021b, Neu et al. 2021, Wang and Mao 2022]* and ODE analysis *[Smith et al. 2021]* communities.

* The discussion around the lower bound on the target domain population risk of Corollary 4.1. is confusing (at least to me). The result says that $R_\mu(w) - \sqrt{2R^2 D_{\textnormal{KL}}(\mu' \lVert \mu)} \leq R_{\mu'}(w)$. The discussion refers to this as a fundamental difficulty in UDA learning, since using the same predictor $f_w$ the population risk in in the target domain cannot be lower than that of the source domain minus a constant depending on the domain difference. However, I understand this as a potential advantage: one can design algorithms that attain lower population risk in the target domain than in the source domain when the domains are sufficiently different.

* The discussion in Section 5.3. can be strengthened. The authors note that the expected cross-entropy loss for each $Z_i$ can be decomposed as (6) $$\mathbb{E}[\ell(f_W(T_i),Y_i)] = H(Y_i|T_i) + \mathbb{E}[D_{\textnormal{KL}}(P_{Y_i|T_i,W} \lVert Q_{Y_i|T_i,W})] - I(W;Y_i|T_i).$$
Then they proceed to say that minimizing the expected cross entropy may not adequately reduce $H(Y_i|T_i)$ but increase $I(W;Y_i|T_i)$. However, note that $I(W;Y_i|T_i) = H(Y_i|T_i) - H(Y_i|T_i,W)$, and therefore (6) can be simplified to $$\mathbb{E}[\ell(f_W(T_i),Y_i)] = H(Y_i|T_i,W) + \mathbb{E}[D_{\textnormal{KL}}(P_{Y_i|T_i,W} \lVert Q_{Y_i|T_i,W})],$$
suggesting that minimizing the expected-cross entropy will minimize $H(Y_i|T_i,W)$. Since conditioning reduces the entropy $H(Y_i|T_i,W) \leq H(Y_i|T_i)$, so minimizing $H(Y_i|T_i,W)$ does not need to minimize $H(Y_i|T_i)$, effectively having an effect of maintaining a large conditional mutual information $I(W;Y_i|T_i)$.

* The proof of Theorem 4.2. seems to need the extra assumption of having symmetric losses, i.e., losses such that $\ell(x,y) = \ell(y,x)$ for all $x,y$.

(*Sorry for the slight change in notation in the following two bullet points, the OpenReview render is not happy with subscripts in the expectation symbols $\mathbb{E}$*.)

* Theorem 4.3. and its proof seem a little confusing to me. In *[Germain et al. 2020]* the domain disagreement is defined as $$\textnormal{dis}(P_X,P_{X'}) = \big| E_{W,W'} \big[E_{X'}[\ell(f_W(X'),f_{W'}(X')] - E_{X}[\ell(f_W(X),f_{W'}(X)] \big] \big|,$$
however in the text it is written as $$\textnormal{dis}(P_X,P_{X'}) = \big| E_{W,W',X'}[\ell(f_W(X'),f_{W'}(X')] - E_{W,W',X}[\ell(f_W(X),f_{W'}(X)] \big|,$$
which is confusing since usually there is a relationship between $W$ and $X$, as the parameters of the model's function depend on the data. Hence, it appears like if in the proof of Theorem 4.3. there is a flaw regarding the conditioning. Let's focus on the first array of three inequalities and on the first term on the right-hand-side. Essentially, the proof states that $$\sup_{W, W' \in \mathcal{W}^2} E_{X'}[t \ell(f_W(X'),f_{W'}(X')] \leq E_{W,W'} \big[E_{X'}[t \ell(f_W(X'),f_{W'}(X')] \big] \leq E_{W,W',X'} [t \ell(f_W(X'),f_{W'}(X')],$$
and if $W$ and $X$ are related this is not true.

    * This may be easily fixed by making sure that the independence of $W,W'$ and $X$ and $X'$ is clear in the definition from *[Germain et al. 2020]*.

* In Theorem B.1. it is stated that without loss of generality one may assume that $\alpha \leq \mathbb{E}[e^{g(\hat{Z})}] \leq \mathbb{E}[e^{g(Z)}]$, where $\hat{Z} \sim \hat{\mu}$ and $Z \sim \mu$, for some constant $\alpha > 0$ and any $g$. I can understand the fist inequality concerning $\alpha$, but I do not see why the moment generating function of $g(Z)$ dominates that of $g(\hat{Z})$ in general.

* In Theorem B.2., after solving for Theorem 11.2.1. in *[Cover and Thomas, 2006]* one gets the standard dependence on the probability $\frac{1}{n} \log(1/\delta)$, not $\frac{1}{n \log \delta}$.


**References**

*[Cover and Thomas, 2006]* Elements of information theory. \
*[Xu and Raginsky, 2017]* Information-theoretic analysis of generalization capability of learning algorithms.\
*[Pensia et al. 2018]* Generalization error bounds for noisy, iterative algorithms. \
*[Bu et al. 2019]* Tightening mutual information based bounds on generalization error. \
*[Negrea et al. 2019]* Information-theoretic generalization bounds for SGLD via data-dependent estimates. \
*[Germain et al. 2020]* PAC-Bayes and domain adaptation. \
*[Haghifam et al. 2020]* Sharpened generalization bounds based on conditional mutual information and an application to noisy, iterative algorithms. \
*[Rodríguez-Gálvez et al. 2021b]* On random subset generalization error bounds
and the stochastic gradient Langevin dynamics algorithm. \
*[Smith et al. 2021]* On the origin of implicit regularization in stochastic gradient descent. \
*[Rodríguez-Gálvez et al. 2021]* Tighter expected generalization error bounds via Wasserstein distance. \
*[Neu et al. 2021]* Information-theoretic generalization bounds for stochastic gradient descent. \
*[Wang and Mao, 2022]* On the generalization of models trained with SGD: information-theoretic bounds and implications.

**Summary Of The Paper:**

This paper considers the problem of *unsupervised domain adaptation* (UDA): described as a supervised learning problem where the source and target domain distributions are different and the learning algorithm has access to labelled source and unlabeled target data.

The paper considers the *empirical-to-population* (EP) generalization error in this setting, which is defined as
$$\textnormal{Err}(w) = R_{\mu'}(w) - R_s(w),$$

where $R_{\mu'}$ is the population risk in the target domain distribution $\mu'$ and $R_s$ is the empirical risk in the source domain with a dataset $s = \lbrace z_i\rbrace_{i=1}^n$ sampled i.i.d. from the source distribution $\mu$.

Traditionally, in the domain adaptation literature, the studied error is the *population-to-population* (PP) generalization error
$$\widetilde{\textnormal{Err}}(w) = R_{\mu'}(w) - R_\mu(w),$$

since one may decompose the EP generalization error $\textnormal{Err}$ into the PP error and the classical generalization error, for which there is a large literature in its characterization, namely
$$
    \textnormal{Err}(w) = \widetilde{\textnormal{Err}}(w) + \big[R_\mu(w) - R_s(w) \big].
$$

The paper employs known techniques from the classical generalization community to bound differences of expectations of the same function to develop bounds of both the EP error and the PP error. These bounds are both tighter and more insightful than previous bounds: e.g., the bounds in Section 5 can make use of the unlabeled data. This way, also taking from the classical generalization community, they find bounds on the stochastic gradient Langevin dynamics algorithm (SGLD).

Then, they use the insights obtained from the derived bounds to develop regularization strategies that result in better EP generalization error in their experiments.

**Summary Of The Review:**

This paper finds upper bounds on the empirical-to-population (EP) and population-to-population (PP) generalization error in unsupervised domain adaptation (UDA).

To do so, the paper employs various techniques used to bound the classical generalization error. Using these techniques the resulting bounds are tighter than other bounds in the literature and some provide interesting insights on what is necessary/important to succeed in UDA.

Finally, the paper uses such insights to develop regularization strategies that result in better domain adaptation than previous schemes.

The paper is a good contribution to the community and therefore I lean towards acceptance. Nonetheless, there are some issues to improve (see Weaknesses) such as proper crediting to the inspiring work on bounding the classical generalization error, from which most of the bounding techniques are taken; the discussion around Corollary 4.1. and (6); the assumptions needed for 4.2.; the point raised about Theorem 4.3.; and some statements in Theorems B.1. and B.2. I believe all of them are small and doable during the rebuttal phase.


**Minor comments and nitpicks that did not impact the score of the review**

* In the third paragraph of the introduction you use the terms PP and EP without explaining them. It would be good to write what they mean.
* In the 4th line of the Preliminary, 'hypothesis space of ~~interesting~~ interest,'
* In page 3, line 5, 'an algorithm $\mathcal{A}$ that takes'
* In Assumption 3: is $\beta$-Lipschitz ~~continues~~ continuous.
* Also in Assumption 3. Lipschitz continuity is defined with respect to a metric $d$. So I recommend to say that $\ell(f_w(X),Y)$ is $\beta$-Lipschitz continuous in $\mathcal{Z}$ with respect to a metric $d$ on $\mathcal{Z}$ for any $w \in \mathcal{W}$.
* You say that (3) is remarkable. What is remarkable about such a result? It seems reasonable and in line to the previous literature.
* In page 5, could you add a reference (or some references) after the claim that "A common method is to assign pseudo labels to target data based on a learned source classifier".
* In Theorem 4.5 and its proof, the Lipschitz constant should always be $\beta$, sometimes you also write $L$.
* There is a change in notation for probabilities in the second to last paragraph in page 8. The one starting with "Notice that...".
* In the proof of Theorem 4.2 one can go from (8) to the equation after (9) directly using the triangle inequality.
* Could you provide the exact equation or theorem for the classic VC-dimenstion generalization bound in the proof of Theorem B.1.
* In the last line of page 18: 'might exist a ~~more optimal~~ faster convergence rate'.
* One can go directly from the third to the last equality in Section C.7. using the classical definition of conditional mutual information $I(X;Y|Z) = H(Y|Z) - H(Y|X,Z)$.

---

> ### Author Response · Authors · 2022-11-16
> **To Reviewer mqeg:**
>
> We would like to sincerely thank you for spending great effort going through the entire paper, including all proof details in the Appendix. We are also grateful for your constructive comments, which have helped us to polish the presentation. Our responses follow.
>
> >- Although the classical generalization error papers introducing the techniques used to bound the difference of expectations are quickly cited in the related work section, it is not clear in the main text where and how these techniques are used or how they inspired the bounds in the main text...
>
> **Response.** Much of our analysis is indeed based on these previous techniques mentioned by the reviewer, we have made our best effort giving credits to these previous works along our development and proofs, despite the space limit. Please see the revised version of the paper.
>
> >- The discussion around the lower bound on the target domain population risk of Corollary 4.1. is confusing (at least to me)...... one can design algorithms that attain lower population risk in the target domain than in the source domain when the domains are sufficiently different.
>
> **Response.** The lower bound here indicates that for all learning algorithms, the target domain population risks must be larger than this value. Thus it suggests an impossibility of learning, i.e., no algorithm can attain a population risk lower than this value. Thus it highlights a fundamental difficulty and is a hardness result.
>
> On the other hand, we agree with your comments that the lower bound may suggest a potential advantage in the cases when the target domain and source domain have significant difference. Indeed in [1], the authors show that learning the target domain can sometimes be easier than learning the source domain. But our lower bound by itself only suggests that such potential advantage **may** exist, without asserting that it must exist. In addition, unlike upper bounds, which often inspire algorithm designs, such a lower bound can hardly be utilized for such a purpose.
>
> >- The discussion in Section 5.3. can be strengthened ... ...
>
> **Response.** Thank you very much for providing this alternative explanation around our Eq. (6). But we feel your explanation may be as strong as ours, not stronger.
> Note that $H(Y_i|T_i, W)$ is a functional of the joint distribution $P_{Y_iT_iW}$ of $(Y_i, T_i, W)$. If $H(Y_i|T_i, W)$ is reduced, the corresponding  $P_{Y_iT_iW}$ will result in a change of $P_{Y_iT_i}$, which in turn changes $H(Y_i|T_i)$. Note that the change can be either an increment or a decrement, its impact on
> $I(W, Y_i|T_i)= H(Y_i|T_i) - H(Y_i|T_i, W)$ could be either increasing or decreasing.
>
> One advantage of our explanation is that it directly suggest that minimizing $\mathbb{E}[\ell(f_W(T_i), Y_i)]+I(W;Y_i|T_i)$ will reduce $H(Y_i|T_i)$, which then naturally inspires a regularization scheme (controlling the label information).
>
> >- The proof of Theorem 4.2. seems to need the extra assumption of having symmetric losses...
>
> **Response.** Thank you for pointing this out. Indeed, if
> $\lambda^*$ is as we defined, then we will need the loss to be symmetric. We have added this condition to Assumption 4 in the revision. Notably, it is in fact to define $\lambda^*$ slightly differently so that the condition of symmetric loss is not needed. But we choose not to elaborate in the paper.
>
> >- Theorem 4.3. and its proof seem a little confusing to me.
>
> **Response.** Agreeably here our expectation notation introduces unnecessary confusion. We indeed use the same definition with [Germain et al. 2020] where $W,W'\sim P_W^{\otimes 2}$. which are independent of $X$ and $X'$. We have corrected the definition of domain disagreement in the revision.
>
> >- In Theorem B.1. it is stated that without loss of generality one may assume that ...... but I do not see why the moment generating function of $g(Z)$ dominates that of
> $g(\hat{Z})$ in general.
>
> **Response.** Our statement is a little sloppy and may confuse readers. We indeed only need that there exists an $\alpha$ s.t. $\alpha\leq \min\\{E_\mu[e^{g(Z)}],E_{\hat{\mu}}[e^{g(Z)}]\\}$, and the proof holds for either $E_\mu[e^{g(Z)}]\leq E_{\hat{\mu}}[e^{g(Z)}]$ or $E_{\hat{\mu}}[e^{g(Z)}]\leq E_{\mu}[e^{g(Z)}]$. We have modified the statement in the revision.
>
> >- In Theorem B.2., after solving for Theorem 11.2.1. in [Cover and Thomas, 2006] one gets the standard dependence on the probability $\frac{1}{n}\log{(1/\delta)}$ not $\frac{1}{n\log{\delta}}$.
>
> **Response.** Thanks for pointing out this typo. We have modified this in the revision.
>
> >- Minor comments ......
>
> **Response.** We again thank you sincerely for providing these useful comments to our paper. We have revised our paper according to these comments.
>
> [1] Hanneke, Steve, and Samory Kpotufe. "On the value of target data in transfer learning." NeurIPS 2019.

---

> > ### Comment · Reviewer_mqeg · 2022-11-21
> > **Answer to response**
> >
> > Thank you for your response. Most of my comments have been addressed. Just a couple points where I have further remarks/clarifications:
> >
> > * *Regarding the discussion around Corollary 4.1:*
> >
> > I still believe that the phrasing around the lower bound is not as helpful as it could be. The reason is that the current text simply states in words what the bound says. But when learning on $\mu$ one already expects that $R_{\mu}(w) \leq R_{\mu'}(w)$, so it is expected that $R_{\mu}(w) - c(\mu',\mu) \leq R_{\mu'}(w)$ for some function $c$. For me, the interesting part of this result is the quantification that $c(\mu',\mu) \geq \sqrt{2 R^2 D_{\textnormal{KL}}(\mu' \lVert \mu)}$. That is, the lower bound can be seen as a fundamental limit on the improvement in the target domain $\mu'$ when learning on the source domain $\mu$, since we can write $R_{\mu}(w) - R_{\mu'}(w) \leq  \sqrt{2 R^2 D_{\textnormal{KL}}(\mu' \lVert \mu)}$. I hope I explained better my intention and proposal for the discussion this time.
> >
> > * *Regarding the discussion in Section 5.3:*
> >
> > I guess that my problem with the explanation is that the "why" of the sentence "Then minimizing the expected cross-entropy loss may not adequately reduce $H(Y_1|T_i)$ cause $I(W;Y_i|T_i)$ to significantly increase, particularly when the model capacity is large." is not given. It was nothing against the decomposition in (6), which indeed suggests later a "controlling the label information" scheme. I would recommend explaining the "why" either as I wrote in my comment or as you did in yours (but in there also include the fact that $H(Y_i|T_i) \geq H(Y_i|T_i,W)$, since that is a reason that minimizing the latter can result in an increase or decrease of the former).
> >
> > It is not a major deal since some readers may understand why directly, but UDA readers without a background in information theory may be wondering why is that sentence true, and a justification can be helpful to them.

---

> > > ### Author Response · Authors · 2022-11-22
> > > **Thank you for your feedback**
> > >
> > > Thanks for your prompt reply.
> > >
> > > >- Regarding the discussion around Corollary 4.1:
> > >
> > > **Response.** We now agree with your comments "the lower bound can be seen as a fundamental limit on the improvement in the target domain $\mu'$ when learning on the source domain $\mu$" (since $R_{\mu'}$ could be smaller than $R_{\mu}$ but their gap should not be arbitrarily large). It seems better to regard the lower bound as a fundamental limitation on the improvement in the target domain rather than a fundamental difficulty in learning the target domain. We will revise our discussion in the next update.
> > >
> > > >- Regarding the discussion in Section 5.3:
> > >
> > > **Response.** We will use our explanation while also following your advice by pointing out the fact $H(Y_i|T_i)>H(Y_i|T_i,W)$ in the next version.
> > >
> > > If you think there is anything else that can be done to improve this paper, we will be very pleased to hear your input.

---

### Official Review · Reviewer_DmXB · 2022-10-19

**Confidence:** 3
**Correctness:** 3
**Technical Novelty And Significance:** 3
**Empirical Novelty And Significance:** 3
**Recommendation:** 8

**Clarity, Quality, Novelty And Reproducibility:**

The paper is well written and the code is provided.


**Strength And Weaknesses:**

Strength:
1、The authors rigorously prove some generalization error bound. One of the bounds is associated only
with the learning algorithm, which is very different from other work.
2、The author designs a new algorithm on the UDA problem based on their theory and achieves good
results compare to other methods in the otatedMNIST and Digits datasets.
3、The author has relaxed the previous strong assumptions[1], so that the conclusions drawn are closer to
the real situation. And the author add gradient penalty based on this, which can be applied to any existing
UDA algorithm
[1] A. Tuan Nguyen, Toan Tran, Yarin Gal, Philip Torr, and Atilim Gunes Baydin. KL guided domain
adaptation. In International Conference on Learning Representations, 2022. URL https:
//openreview.net/forum?id=0JzqUlIVVDd.

weakness：

1、The experiments are limited. Experiments on some popular datasets ( Office-Home, Office-31) have not been reported in the paper. The authors do not state how the results of the experiments match the theory analyzed clearly.

2、I don't understand how the first inequality sign of Equation (14) is derived from Lemma 1. How to use the independence condition between the algorithms  and unseen target data.

3、Compare to other UDA paper, this paper lack visual instructions such as t-SNE.

4、The paper lacks ablation experiments with gradient penalties to demonstrate the effectiveness of this design.

**Summary Of The Paper:**

This paper use an information-theoretic framework to analyze the generalization ability of hypotheses and
learning algorithm of unsupervised domain adapation problem.

**Summary Of The Review:**

This paper give some bounds of UDA problem. The authors propose generalization error bounds relevant only to the learning algorithm, which is very different from previous work. In addition, the authors analyze previous work and relax its strong assumptions. And based on this, a controlling label information algorithm is proposed. However, there are some details of the proof that are not clearly written. Besides, there is a lack of explanation of experimental results and theoretical analysis.

---

> ### Author Response · Authors · 2022-11-16
> **To Reviewer DmXB:**
>
> Thank you for your constructive comments. Our responses follow.
>
> >- The experiments are limited. Experiments on some popular datasets ( Office-Home, Office-31) have not been reported in the paper. The authors do not state how the results of the experiments match the theory analyzed clearly.
>
> **Response.** We have added additional experimental results on the VisDA17 dataset in Appendix. Time permitting, we will try to include more experiments.
>
> Regarding the empirical results and theoretical results, we would like to note that the design of our regularization techniques is completely motivated by our theoretical analysis (e.g., GP by Theorem 5.3). When GP or CL combined with ERM/KL guided algorithm,  we can see that the performance is further boosted, which justifies that penalizing gradients or restricting the memorization of labels will indeed improve the generalization performance, as suggested by the analysis.
>
> >- I don't understand how the first inequality sign of Equation (14) is derived from Lemma 1. How to use the independence condition between the algorithms and unseen target data.
>
> **Response.** From Lemma A.1, we are suppose to have
> $$
> \mathrm{D\_{KL}}(P\_{W,Z\_i|X\_j'=x\_j'}||P\_{W|X\_j'=x\_j'}P\_{Z'})=\sup\_{g\in\mathcal{G}}\mathbb{E}\_{P\_{W,Z\_i|X\_j'=x\_j'}}{g(W,Z\_i)}-\log{\mathbb{E}\_{P\_{W|X\_j'=x\_j'}P\_{Z'}}{\exp{g(W,Z')}}},
> $$
> where $\mathcal{G}=\\{g:\mathcal{W}\times\mathcal{Z}\rightarrow \mathbb{R} \text{ s.t. } g(W,Z)< \infty\\}$. Now we choose a specific form of function $g$, i.e. $g=t\ell(f_W(X_i),Y_i)$ for any $t\in \mathbb{R}$. That is to say, while $\mathcal{G}$ contains all the measurable function of $(W,Z)$, we now only consider a subset of $\mathcal{G}$ which is  parameterized by $t$. Thus,
> $$
> \sup\_{g\in\mathcal{G}}\mathbb{E}\_{P\_{W,Z_i|X_j'=x_j'}}{g(W,Z_i)}-\log{\mathbb{E}\_{P_{W|X_j'=x_j'}P_{Z'}}{\exp{g(W,Z')}}}\geq\sup\_t \mathbb{E}\_{P_{W,Z_i|X_j'=x_j'}}{t\ell(f_W(X_i),Y_i)}-\log{\mathbb{E}\_{P_{W|X_j'=x_j'}P_{Z'}}{\exp{t\ell(f_W(X'),Y')}}}.
> $$
>
> The independence condition between the algorithms and unseen target data is used to obtain Eq.(14):
> $$
> \mathrm{D_{KL}}\left(P_{W,Z_i|X_j'=x_j'}||P_{W,Z'|X_j'=x_j'}\right) =\mathrm{D_{KL}}\left(P_{W,Z_i|X_j'=x_j'}||P_{W|X_j'=x_j'}P_{Z'}\right).
> $$
> This is because $P_{W,Z'|X_j'=x_j'}=P_{Z'|X_j'=x_j',W}P_{W|X_j'=x_j'}$ and $P_{Z'|X_j'=x_j',W}=P_{Z'}$ (namely the unseen/testing target data $Z'$ is independent of the unlabelled training data $X_j'$ and the algorithm output $W$).
>
> >- Compare to other UDA paper, this paper lack visual instructions such as t-SNE.
>
> **Response.** Thanks for the suggestion, and we have added the visualization results based on t-SNE in Appendix D.6 of the revised paper.
>
> >- The paper lacks ablation experiments with gradient penalties to demonstrate the effectiveness of this design.
>
> **Response.** The ablation study on the effect of gradient penalty was added to the revision (see Table 3 in Appendix D.5). Indeed, we also note that by comparing the results of KL and KL-GP in Table 1, the effectiveness of adding gradient penalties has been already demonstrated.

---

### Official Review · Reviewer_DzXY · 2022-10-24

**Confidence:** 3
**Correctness:** 3
**Technical Novelty And Significance:** 4
**Empirical Novelty And Significance:** 4
**Recommendation:** 8

**Clarity, Quality, Novelty And Reproducibility:**

Paper is well written and not difficult to follow. The present generalization error bound, especially Theorem 5.1 and obtained algorithms seems novel.


**Strength And Weaknesses:**

# Strength
-  The authors discuss how the obtained bound relates to existing algorithms in detail. The authors discussed how the KL divergence obtained by the change-of-measure inequality relates to margin alignment methods in Theorem 4.2 and Sec 5.3.
- The second generalization error bound (expected EP generalization error) in Theorem 5.1 seems novel and leads to the useful algorithm in Theorem 5.3.
- The obtained algorithm is solid based on Theorem 5.3 and seems easy to implement and improve existing methods, as shown in Table 1.

# Questions and Weakness
- According to the Appendix, the authors used cross-entropy loss, but I am not sure the cross-entropy loss satisfies Assumption 2 of subgaussianity since the cross-entropy loss is unbounded.
- In the algorithm, when taking the gradient of the gradient penalty, does the Hessian appears in the update rule? Or did the authors use detach function in pytorch so that the Hessian would not appear?
- The method of controlling label information requires additional memory and computation since it requires the subnetwork to generate $\tilde{W}$.
- I would like to know the exact definition of $P_{W,Z'|X'_j=x'_j}$. Does this mean $Z'=(Y',X')$ is the test data in the target domain, and $X'_j=x_j'$ is the training data of unlabelled data in the target domain ?
- I would like to know the performance of CL+GP, where both the controlling label information and gradient penalty is applied. Does this lead to better performance? Or does the use of two techniques regularize the information too much and is harmful to UDA?

**Summary Of The Paper:**

This work proposed the information-theoretic generalization error analysis of UDA. Especially the authors provided two approaches, PP generalization error and expected EP generalization error, which are based on the change-of-measure inequalities. Then the authors discussed these bounds about existing algorithms, especially margin alignment. Finally, based on the theoretical analysis, the authors proposed two algorithms: gradient regularization based on mutual information and controlling label information based on the entropy decomposition. The numerical experiments support the usefulness of these algorithms.

**Summary Of The Review:**

This paper presents the novel information-theoretic generalization error bound in UDA, which significantly generalizes the existing analysis and algorithms.
Also, the authors showed that the gradient penalty is useful in UDA by connecting it to the obtained MI in the generalization error bound. I think the results obtained in this paper are worth publication.

---

> ### Author Response · Authors · 2022-11-16
> **To Reviewer DzXY:**
>
> Thank you very much for your positive comments. Below we discuss your comments and concerns in detail.
>
> >- According to the Appendix, the authors used cross-entropy loss, but I am not sure the cross-entropy loss satisfies Assumption 2 of subgaussianity since the cross-entropy loss is unbounded.
>
> **Response.** Indeed the cross-entropy loss is used in our training. Albeit the unboundedness of cross-entropy loss, we believe that it indeed has a sub-Gaussian distribution under the SGD training since the loss remains bounded (and decreasing) during training. We have not made an effort in rigorously proving the subGaussianality of the loss. Such an effort would require a careful examination of the SGD dynamics and involve significant technicality which seems to only serve as a digression from the main thrust of the paper. In practice, to estimate the variance proxy of the loss when treating it as a sub-Gaussian random variable, one can take the maximum seen value of the loss as its upper bound, and use this bound to obtain an estimate of the variance proxy. But we believe that such an estimate is already a significant over-estimate.
>
> > In the algorithm, when taking the gradient of the gradient penalty, does the Hessian appears in the update rule? Or did the authors use detach function in pytorch so that the Hessian would not appear?
>
> **Response.** The current implementation indeed includes the direct computation of Hessian-vector products which is inspired by the work of [A]. This can be easily improved by the finite-differences approximation technique as introduced in [B], which is able to avoid computing Hessian but requires one additional forward-backward pass.
>
> >- The method of controlling label information requires additional memory and computation since it requires the subnetwork to generate $\widetilde{W}$
>
> **Response.** Yes, it does need additional memory. However, the subnetwork is just an additional classifier which is normally a one-layer network in most cases.
>
> >- I would like to know the exact definition of $P_{W,Z'|X_j'=x_j'}$. Does this mean $Z'=(Y',X')$ is the test data in the target domain, and $X_j'=x_j'$ is the training data of unlabelled data in the target domain ?
>
> **Response.** The reviewer's understanding is correct. Here $Z'$ is the unseen test data independently drawn from the target domain and $X_j'$ is the known unlabelled training target data. Since $Z'$ is independent of both $X_j'$ and $W$, $P_{W,Z'|X_j'=x_j'}$ is just equal to $P_{W|X_j'=x_j'}P_{Z'}$.
>
> >- I would like to know the performance of CL+GP, where both the controlling label information and gradient penalty is applied. Does this lead to better performance? Or does the use of two techniques regularize the information too much and is harmful to UDA?
>
> **Response.** We indeed tried CL+GP before. Combining these two schemes will not be harmful to UDA but the performance is very close to using GP alone, and given that CL+GP introduces more computation complexity (since we may also need to penalize the gradients of the additional auxiliary subnetwork), we did not report its performance.
>
> [A] Nagarajan, Vaishnavh, and J. Zico Kolter. "Gradient descent GAN optimization is locally stable." NeurIPS 2017.
>
> [B] Geiping, Jonas, et al. "Stochastic Training is Not Necessary for Generalization." ICLR 2022.

---

### Official Review · Reviewer_jaro · 2022-10-26

**Confidence:** 4
**Clarity, Quality, Novelty And Reproducibility:** Some details and intuitions need more…
**Correctness:** 2
**Technical Novelty And Significance:** 3
**Empirical Novelty And Significance:** 2
**Recommendation:** 3

**Strength And Weaknesses:**

Strength:

1. The theoretical analysis in Section 5 is interesting and novel to me.

2. Applying SGLD to UDA, though itself is not initially developed in this paper, is novel to me.


Weakness:

1. As the authors claimed, the KL term upper bounds some other measures (e.g., Wasserstein distance), which indicates that the bound is looser and therefore less informative.

2. In the literature, algorithm-dependent bounds have already been developed. For example, in [1], the expected target loss is upper bounded in terms of the performance gap, which depends on the learning algorithm. Unfortunately, the authors did not provide an in-depth analysis of or comparison with existing works along this line, which is related to this paper.

3. Page 5: squared loss does not satisfy the triangle inequality. In fact, it satisfies the \emph{approximate triangle inequalities} [2].

4. The entire theoretical analysis in Section 4 is incremental – it is more like a rephrasing of existing conclusions using the information-theoretic language, which does not provide any new insight. For example, it is hard to understand what is the difference between $\lambda^*$ in Theorem and $\lambda$ in [3].

5. More detailed description is needed to understand the intuition behind $I(W;Z|X_j’)$ or $I(X_j’;Z|W)$.

6. The authors noted that the bound in Theorem 5.3 only depends on $n$. From my aspect, it indicates that the bound cannot verify the benefits of target data, but just SGLD itself. In other words, there is a gap between Theorem 5.1 and Theorem 5.3.

7. The empirical results are not convincing. The authors should evaluate the method on more practical benchmark data sets (e.g., Office-Home, VisDA, DomainNet…). In addition, compared with a recent baseline (KL), the improvement is not significant.


[1] Wang, B., Mendez, J., Cai, M. and Eaton, E., 2019. Transfer learning via minimizing the performance gap between domains. Advances in Neural Information Processing Systems, 32.

[2] Crammer, K., Kearns, M., & Wortman, J. (2008). Learning from Multiple Sources. Journal of Machine Learning Research, 9(8).

[3] Ben-David, S., Blitzer, J., Crammer, K., Kulesza, A., Pereira, F., & Vaughan, J. W. (2010). A theory of learning from different domains. Machine learning, 79(1), 151-175.


Questions:

1. On page 5, is the random variable T’ above Eq. 4 sampled from the source or target?

2. Remark 5.1, why the mutual information will vanish as $n, m \rightarrow \infty$ (as it is already defined over the expectation)?

3. Theorem 5.1, the average over $m$ and $n$ seem unnecessary since RHS is already defined over the expectation.

4. The mathematical definition of $I^{T_j’}(W;Y_i|T_i)$ is not clear to me.

5. Theorem 5.3 is strange to me -- does it mean a small T will give a better generalization performance?






**Summary Of The Paper:**

This work provides UDA bounds based on the information-theoretic tools. Accordingly, new UDA methods are proposed with empirical evidence.

**Summary Of The Review:**

While this paper provides some new insights into UDA, I don't think it is ready to publish in a top-tier conference (e.g., ICLR) given the weaknesses mentioned. However, it could be a good paper if the authors can carefully address the issues.

---

> ### Author Response · Authors · 2022-11-16
> **To Reviewer jaro:**
>
> We thank you sincerely for your comments to our paper. Our responses follow.
>
> >- As the authors claimed, the KL term upper bounds some other measures (e.g., Wasserstein distance), which indicates that the bound is looser and therefore less informative.
>
> **Response.** First we would like to note that it may not be fair to conclude that our KL-based bounds are looser than those based on other domain discrepancy measures. This is because our KL bounds require weaker assumptions (i.e., sub-Gaussianality). Under such assumptions, those other bounds do not even hold. Only under some additional assumption (e.g., bounded loss), our KL bounds become the upper bound of the other bounds (e.g. Total-variation based bound in Corollary 4.3).  On the other hand, despite the importance of having a  tighter bound, we argue that such importance is tampered if the compared bounds are much looser than the true generalization error. From a practical point of view, the usefulness of a bound is arguably more akin to whether it correctly tracks the trend of the true error and whether it may be effectively estimated and controlled in a learning algorithm. From this view point and as demonstrated in our experiments, the KL based bounds is arguably more useful than some other bounds.
>
> >- In the literature, algorithm-dependent bounds have already been developed. For example, in [1], the expected target loss is upper bounded in terms of the performance gap, which depends on the learning algorithm. Unfortunately, the authors did not provide an in-depth analysis of or comparison with existing works along this line, which is related to this paper.
>
> **Response.** Thanks for pointing out [1], which we missed in this submission, and we have added it in the revision as a related work. However, it is worth noting that although [1] and our work both provide algorithm-dependent bounds, they cannot be directly compared. This is because the settings in the two works are different, for example, [1] considers that the data in the target domain are labelled, which is crucial for defining the performance gap in their paper. On the other hand, our work focuses on the UDA setting where the target-domain data used in training are unlabelled. Additionally, results in [1] require that the loss be Lipschitz-continuous and convex, whereas our bounds only require a weaker condition, namely, sub-gaussian loss. Finally, the bounds in [1] are high-probability bounds (e.g., Rademacher complexity bound), while our bounds are for the expected generalization error.
>
> >- Page 5: squared loss does not satisfy the triangle inequality. In fact, it satisfies the \emph{approximate triangle inequalities} [2].
>
> **Response.** Thank you for pointing this out. We stand corrected. Inspired by your criticism and comments concerning approximate triangle inequality, we have presented another theorem generalizing Theorem 4.2 to a version in which the loss function satisfies $\alpha$-triangle inequality. It can be found in the Appendix B.10 of the revision (Theorem B.3), where we also mention that squared loss satisfies $2$-triangle inequality, to which the theorem applies.
>
> >- The entire theoretical analysis in Section 4 is incremental – it is more like a rephrasing of existing conclusions using the information-theoretic language, which does not provide any new insight. For example, it is hard to understand what is the difference between $\lambda^*$ in Theorem and $\lambda$ in [3].
>
> **Response.** Agreeably some analyses in our Section 4 are inspired by many previous works (and $\lambda^*$ is indeed the same with $\lambda$ in [3]). We argue however that Section 4 aims at unifying these previous analyses in the information-theoretic framework. This not only demonstrates the power of information-theoretic tools in analyzing domain adaptation or related learning problems, it also suggests that minimizing KL divergence necessarily regularizes other domain discrepancy measures such as total variation distance/1-Wasserstein distance and domain disagreement. One consequence of this is that it also explains the superior effectiveness of KL guided marginal alignment methods to other methods in practice.
>
> >- More detailed description is needed to understand the intuition behind $I(W;Z|X_j')$ or $I(Z;X_j'|W)$.
>
> **Response.** The term $I(W;Z_i|X_j')$ measures the dependence of output $W$ on a single source domain example $Z_i$ upon observing the target-domain (unlabelled) example $X_j'$. This quantity upper bounds the first term in Theorem 5.1 (more precisely $\mathbb{E}_{X_j'}\sqrt{I^{X_j'}(W;Z_i)}\leq \sqrt{I(W;Z_i|X_j')}$). When $I(W;Z_i|X_j')$ is small, say, close to 0, then $W$ is nearly independent of the training example $Z_i$ in the source domain upon observing target domain example $X_j'$. This will allow good generalization of $W$ to unseen examples in the source domain.

---

> > ### Author Response · Authors · 2022-11-16
> > **To Reviewer jaro (cont.):**
> >
> > On the other hand, $I(Z_i;X_j'|W)$ quantifies the gap between $I(W;Z_i)$ and $I(W;Z_i|X_j')$. We hope $I(Z_i;X_j'|W)$ is small enough so that knowing $X_j'$ will not increase the dependence between $Z_i$ and $W$.
> >
> > >- The authors noted that the bound in Theorem 5.3 only depends on $n$. From my aspect, it indicates that the bound cannot verify the benefits of target data, but just SGLD itself. In other words, there is a gap between Theorem 5.1 and Theorem 5.3.
> >
> > **Response.** We first note that we also remark on this in the last paragraph of Section 5.2 in the original submission.
> >
> > Furthermore, Theorem 5.3 is indeed looser than Theorem 5.1. This gap is mainly due to two factors:
> > 1. A term $\frac{1}{m}\sum_{j=1}^m\mathbb{E}\sqrt{I^{X_j'}(W; Z_i)}$ is relaxed by $\sqrt{I(W; Z_i|S_{X'}')}$, which removes the dependency of the final bound on $m$, the target-domain sample size.
> > 2. Unrolling the mutual information $I^{X_j'}(W; Z_i)$ in Theorem 5.1 via chain rule and data-processing inequality.
> >
> > It's also possible to let $\frac{1}{m}\sum_{j=1}^m$ explicitly appear in the gradient norm based bound if the sampling rule in SGD to form mini-batches is considered more carefully (i.e., with or without replacement sampling). This would reduce the gap due the second factor. Note that different sampling rules will give the different Markov chains and introduce additional complexity to the application of data processing inequality). The current Theorem 5.3 is applicable to any sampling rule.
> >
> > Notably our Theorem 5.3 mainly aims to provide a theoretical justification for the gradient penalty term, and demonstrate that even in the ERM algorithm (which only uses source domain data), this regularization term can non-trivially boost the performance (as shown in Table 1). For this reason, we made no attempt to include the dependency of $m$ in the bound.
> >
> > >- The empirical results are not convincing. The authors should evaluate the method on more practical benchmark data sets (e.g., Office-Home, VisDA, DomainNet…). In addition, compared with a recent baseline (KL), the improvement is not significant.
> >
> > **Response.** We have added the experimental results on VisDA17 in the Appendix D.7 of the revision.
> >
> > In the Rotated MNIST dataset, given that KL-GP obtains 91.2% testing accuracy while KL has 86.4% testing accuracy, we argue that the performance improvement should not be regarded as insignificant. In the other two datasets experimented (i.e. Digits and VisDA17), the two regularization techniques can still boost the performance of KL, although the improvement is not as significant as in Rotated MNIST. We note that the regularization techniques can benefit the generalization performance but may not change the fundamental limits of the algorithm, e.g., using marginal alignment alone may not perform well enough on VisDA17, and some conditional alignment techniques are needed.
> >
> > >- On page 5, is the random variable T’ above Eq. 4 sampled from the source or target?
> >
> > **Response.** It is sampled from the target domain and we have explicitly noted this in the revision.
> >
> >  >- Remark 5.1, why the mutual information will vanish as $n,m\rightarrow \infty$ (as it is already defined over the expectation)?
> >
> > **Response.**  Note that $S$ depends on $S_{X'}'$ given $W$, so intuitively the dependence between each  individual instance $Z_i$ and $X_j'$ is weaker when $n$ and $m$ become larger. More precisely, W.L.O.G let $i=j=1$, and recall that $W=\mathcal{A}(S_{X'}',S)$, when $n,m\rightarrow \infty$, taking $S$ and $S_{X'}'$ as the input of the algorithm is nearly equivalent to computing $W$ based on the source distribution $\mu$ and the target distribution $P_{X'}$, thus, $W$ will only depend on the two distributions, without depending on the realizations $Z_1$ and $X'_1$ drawn respectively from the two distributions, that is,  $I(Z\_1;X\_1'|W)=I(Z\_1;X\_1'|\mathcal{A}(\mu,P\_{X'}))=I(Z\_1;X\_1')=0$.
> >
> > In the other extreme, if $n=1$ and $m=1$, then $W=\mathcal{A}(X_1',Z_1)$, and the quantity $I(Z_1;X_1'|\mathcal{A}(X_1',Z_1))$ should be large. When $n$ and $m$ increase, it becomes $I(Z_1;X_1'|\mathcal{A}(X_{1:m}',Z_{1:n}))$. Now we want to guess $Z_1$ from $X_1'$, this should be easier when having the knowledge of $\mathcal{A}(X_1';Z_1)$ compared with when having the knowledge of $\mathcal{A}(X_{1:m}';Z_{1:n})$.
> >
> > >- Theorem 5.1, the average over $m$ and $n$ seem unnecessary since RHS is already defined over the expectation.
> >
> > **Response.** They can not be simply removed. Removing
> > $\frac{1}{mn}\sum^{m}\_{j=1}\sum^{n}\_{i=1}$ requires
> >  the algorithm to satisfy some symmetric property, namely, that the output does not depend on the ordering of the points in the input. This would require that $P_{W|Z_i,X_j'}$ be invariant w.r.t. $i,j$. This condition may not hold for all the algorithms.

---

> > > ### Author Response · Authors · 2022-11-16
> > > **To Reviewer jaro (cont.):**
> > >
> > > >- The mathematical definition of $I^{T_j'}(W;Y_i|T_i)$ is not clear to me.
> > >
> > > **Response.** Recall that $T_i$ is the representation of the $i$th source sample and $T_j'=t_j'$ is a specific realization of the $j$th target sample's representation. Your confusion may arise from the fact that we follow the notations in disintegrated mutual information [B].
> > > To be more precise, $I^{T\_j'}(W;Y\_i|T\_i)$ without  expectation should be understood as $I^{T\_j'=t\_j'}(W; Y\_i|T\_i)$ or $I(W; Y\_i|T\_i, T\_j'=t\_j')$ defined as $\mathbb{E}\_{P\_{T\_i|T\_j'=t\_j'}}\mathrm{D\_{KL}}(P\_{W,Y\_i|T\_i,T\_j'=t\_j'}||P\_{W|T\_i,T\_j'=t\_j'}\otimes P\_{Y\_i|T\_i,T\_j'=t\_j'})$. The notation ${\mathbb E}\_{T\_j'}I^{T\_j'}(W;Y\_i|T\_i)$ should be understood as ${\mathbb E}_{t\_j' \sim P\_{T\_j'}}I^{T\_j'=t\_j'}(W; Y\_i|T\_i)$. Then $I^{T\_j'}(W;Y\_i|T\_i)$ is indeed regarded as a function of random variable $T_j'$ and $\mathbb{E}\_{T\_j'}I^{T\_j'}(W;Y\_i|T\_i)=I(W;Y\_i|T\_i,T\_j')$.
> > >
> > > >- Theorem 5.3 is strange to me -- does it mean a small T will give a better generalization performance?
> > >
> > > **Response.** If the empirical risk is sufficiently small with small number $T$ of training iterations, then Theorem 5.3 does suggest that a small $T$ can give a good testing performance. This is consistent with some previous theoretical works in supervised learning such as [A], which argues that "train faster, generalize better".
> > >
> > > [A] Hardt, Moritz, Ben Recht, and Yoram Singer. "Train faster, generalize better: Stability of stochastic gradient descent." ICML 2016.
> > >
> > > [B] Negrea, Jeffrey, et al. "Information-theoretic generalization bounds for SGLD via data-dependent estimates." NeurIPS 2019.

---

> > ### Comment · Reviewer_jaro · 2022-12-07
> > **Some of my concerns remain**
> >
> > First of all, I would like to thank the authors for their detailed responses and apologize for my late response due to some personal business. I have read through the responses carefully but some of my concerns remain.
> >
> > **1. comments on KL divergence:** I agree that KL bounds require weaker assumptions and lead to looser bounds, and some bounds may not even hold under stronger assumptions -- this is kind of common sense in theoretical machine learning -- it will be a breakthrough if someone can provide a tighter bound with weaker assumptions. My point is that a looser bound is usually less informative. The authors have argued that *the usefulness of a bound is arguably more akin to whether it correctly tracks the trend of the true error and whether it may be effectively estimated and controlled in a learning algorithm*. Can it be more specific? More importantly, I do think the authors should emphasize this point in the revised submission.
> >
> > **2. The theoretical analysis in Section 4:** I understand the argument made by the authors, but I still think it is incremental given the existing results. Personally, I think the bound is more *unified* as it is looser.  On the other hand, our viewpoint can be subjective, and I will leave this point to during the discussion with other reviewers.
> >
> > **3. The intuition behind $I(W;Z|X_j')$ or $I(Z;X_j'|W)$** Thanks for the explanation. Did you add it to your revised draft?
> >
> > **4. The bound in Theorem 5.3 only depends on $n$** The authors' response somewhat reflects my concern: Theorem 5.3 may reveal some benefits of SGLD, but has nothing to do with UDA itself.
> >
> > **5. Results on VisDA:** The results are not quite convincing. The SOTA on this data set is usually based on Resnet-101 and is much higher than 72% [1]. I understand that the backbone adopted in this work follows [2] and is Resnet-50, but 72% is still not satisfactory (e.g., [3]). I noted that the authors attribute it to the limitations of marginal alignment, but MDD [3] also only aligns the marginal distributions between domains.
> >
> >
> > [1] https://paperswithcode.com/sota/domain-adaptation-on-visda2017
> >
> > [2]. Nguyen, A. Tuan, et al. "Kl guided domain adaptation." arXiv preprint arXiv:2106.07780 (2021).
> >
> > [3] Zhang, Yuchen, et al. "Bridging theory and algorithm for domain adaptation." International Conference on Machine Learning. PMLR, 2019.

---

> > > ### Author Response · Authors · 2022-12-08
> > > **Thanks for the reply:**
> > >
> > > Thank you for your reply. Our responses follow.
> > >
> > > >1. comments on KL divergence: I agree that KL bounds require weaker assumptions and lead to looser bounds, and some bounds may not even hold under stronger assumptions -- this is kind of common sense in theoretical machine learning -- it will be a breakthrough if someone can provide a tighter bound with weaker assumptions. My point is that a looser bound is usually less informative. The authors have argued that the usefulness of a bound is arguably more akin to whether it correctly tracks the trend of the true error and whether it may be effectively estimated and controlled in a learning algorithm. Can it be more specific? More importantly, I do think the authors should emphasize this point in the revised submission.
> > >
> > > **Response.** This opinion is in fact shared by some researchers in the community. For example, as argued in [1,2], if a looser quantity controls generalization error to a sufficient extent, then this looser quantity may provide a more general explanation of the **empirical generalization** phenomena, as the adequacy of any formally tighter bound is then tautological. Thus, when attempting to explain and improve generalization, a loose bound that is easy to estimate and control during training may suffice. For example, our Theorem 5.3 is looser than Theorem 5.1, but computing gradient norm is significantly easier than estimating the mutual information. Although Theorem 5.1 is more informative, Theorem 5.3 may have more practical advantage.
> > >
> > > On the other hand, a tight bound can be less useful than a loose bound.  We plot the dynamics of scaled Jeffery's divergence (which is the main component of the bound in Corollary 4.2) and target domain testing (0-1) errors in Figure 3(a) in Appendix. We can see that Jeffery's divergence has very similar trend with the testing error. Notably, the Jeffery's divergence, when unscaled, in most cases has a value greater than 1. Then the constant value 1 can be seen as a tighter bound of testing error than the unscaled Jeffery's divergence, but clearly it does not track the trend of the true error and is therefore useless. Furthermore, the true testing error is trivially the tightest upper bound of the testing error, but it is notoriously difficult to estimate, forcing us to find bounding techniques.
> > >
> > > Thank you for advising us to emphasize this point, which we will do in our next revision.
> > >
> > > >2. The theoretical analysis in Section 4: I understand the argument made by the authors, but I still think it is incremental given the existing results. Personally, I think the is more unified as it is looser. On the other hand, our viewpoint can be subjective, and I will leave this point to during the discussion with other reviewers.
> > >
> > > **Response.** Thanks for your understanding; we thus have no further comment to add. It is our hope that this matter could be resolved during the Reviewers-AC discussion period. As our voice in this discussion and for the convenience of all reviewers and AC, we here re-state our previous statement:
> > > "Agreeably some analyses in our Section 4 are inspired by many previous works (and $\lambda^*$ is indeed the same with $\lambda$ in [5]). We argue however that Section 4 aims at unifying these previous analyses in the information-theoretic framework. This not only demonstrates the power of information-theoretic tools in analyzing domain adaptation or related learning problems, it also suggests that minimizing KL divergence necessarily regularizes other domain discrepancy measures such as total variation distance/1-Wasserstein distance and domain disagreement. One consequence of this is that it also explains the superior effectiveness of KL guided marginal alignment methods to other methods in practice."
> > >
> > > >3. The intuition behind $I(W;Z_i|X_j')$ or $I(Z_i;X_j'|W)$ Thanks for the explanation. Did you add it to your revised draft?
> > >
> > > **Response.** Yes, we have added the explanation in the Appendix (see the highlight part in page 22).
> > >
> > > > 4. The bound in Theorem 5.3 only depends on  The authors' response somewhat reflects my concern: Theorem 5.3 may reveal some benefits of SGLD, but has nothing to do with UDA itself.
> > >
> > > **Response.** We respectfully disagree with your comment that Theorem 5.3 has nothing to do with UDA itself. Although Theorem 5.3 does not explicitly depend on the number of target data since we relax $\frac{1}{m}\sum_{j=1}^m\mathbb{E}\sqrt{I^{X_j'}(W; Z_i)}$ by $\sqrt{I(W; Z_i|S_{X'}')}$, the term $I(W; Z_i|S_{X'}')$ still depends on the target sample. Recall that the main component of the bound in Theorem 5.3 is $\sum\_{t=1}^T \mathbb{E}\_{S'\_{X'},W\_{t-1},S}||G\_t-\mathbb{E}\_{Z\_{B\_t}}[G\_t]||^2$, in which $G_t$ and $Z_{B_t}$ both contain the information of unlabelled target data and the target sample $S'_{X'}$ is expected over.

---

> > > > ### Author Response · Authors · 2022-12-08
> > > > **Continue. Sorry for the long reply.**
> > > >
> > > > > 5. Results on VisDA: The results are not quite convincing. The SOTA on this data set is usually based on Resnet-101 and is much higher than 72% [1]. I understand that the backbone adopted in this work follows [2] and is Resnet-50, but 72% is still not satisfactory (e.g., [3]). I noted that the authors attribute it to the limitations of marginal alignment, but MDD [3] also only aligns the marginal distributions between domains.
> > > >
> > > >
> > > > **Response.** We first like to note that the official code and the paper of [3] do not explicitly provide the hyper-parameter settings on VisDA. As a consequence, reproducing their results on VisDA (namely, an accuracy of 74.6%) appears difficult, and this difficulty was also reported by other researchers (please see issues in their github repository [4]).
> > > >
> > > > Nonetheless we wish to note that MDD may indeed have some advantages over the compared methods in our paper. In our paper, the marginal distributions are aligned in the representation space based on their KL divergence without any reference to the classifier. In contrast, MDD minimizes the domain discrepancy induced by a (max-margin) classifier or scoring function, which requires the classifier to provide the pseudo-labels for the target data. In other words, MDD aligns the marginal distributions using a discrepancy measure (resembling the theoretical notion $H\Delta H$ divergence [5]) that explicitly depends on the class of classifiers prescribed by the neural network model.
> > > >
> > > >
> > > > [1] Dziugaite, Gintare Karolina, et al. "In search of robust measures of generalization." NeurIPS 2020.
> > > >
> > > > [2] Haghifam, Mahdi, et al. "Understanding Generalization via Leave-One-Out Conditional Mutual Information." ISIT 2022.
> > > >
> > > > [3] Zhang, Yuchen, et al. "Bridging theory and algorithm for domain adaptation." ICML 2019.
> > > >
> > > > [4] https://github.com/thuml/MDD/issues/3
> > > >
> > > > [5] Ben-David, Shai, et al. "A theory of learning from different domains." Machine learning 79.1 (2010): 151-175.

---

> ### Comment · Area_Chair_EDVK · 2022-11-25
> **Please update your review by today**
>
> Hi,
>
> Since your rating is at very low end and authors provide detailed responses.
> May I know whether these responses change your mind? If yes, please update your rating score accordinlgy.
> If no, please state your concerns ASAP?
>
> Best,
> AC

---

### Comment · Area_Chair_EDVK · 2022-11-20
**Please update your reviews**

Dear Reviewers,

Please make sure that your reviews acknowledge authors’ responses and reflect your current evaluation of the paper. This is particularly important if you didn’t directly engage with the authors during the discussion phase (so the authors don’t know if their response changed your evaluation) or if you expressed an intention to update your rating but did not do so.

Cheers,
AC

---

### Decision · Program_Chairs · 2023-01-20

**Decision:**

Accept: poster

**Justification For Why Not Higher Score:**

1) A reviewer mentioned that the bound is loss and theoretical analysis is kind of incremental given the existing results. 2) The cross-entropy loss, which is used in the paper, might not satisfy the assumption of subgaussianity. 3) It seems that the practical insight provided by the theoretical results is only about restricting the gradient norm, which has nothing to do with UDA itself. 4) The empirical results are not quite convincingIt is not well addressed in the rebuttal. It is not well addressed in the rebuttal.


**Justification For Why Not Lower Score:**

All the reviewers agreed on manuscript novelty, specifically the application of Stochastic Gradient Langevin Dynamics (SGLD) to Unsupervised Domain Adaptation (UDA).

**Metareview: Summary, Strengths And Weaknesses:**

The paper presents work on the upper bounds of UDA problem, in particular the generalization error bounds relevant only to the learning algorithm and also provides techniques for improving generalization with experimental validation.

Strengths
All the reviewers agreed on manuscript novelty, specifically the application of Stochastic Gradient Langevin Dynamics (SGLD) to Unsupervised Domain Adaptation (UDA). It was observed that the work demonstrated tighter upper bounds for two notions of generalization errors compared to previous work. Their work also provides new insights into algorithm designs for the UDA problem.

Weakness:
A reviewer mentioned that the bound is loss and theoretical analysis is kind of incremental given the existing results. The cross-entropy loss, which is used in the paper, might not satisfy the assumption of subgaussianity. It is not well addressed in the rebuttal.  Perhaps some work might have been done to demonstrate subgaussianity since it is explicitly mentioned in the paper.  Another weakness is that the technique is not new which also pointed by reviewers. It seems that the practical insight provided by the theoretical results is only about restricting the gradient norm, which has nothing to do with UDA itself. All these concerns are not well addressed in the rebuttal.



**Note From Pc:**

if the above contains the word "oral" or "spotlight" please see: "oral" presentation means -> notable-top-5% and "spotlight" means -> notable-top-25%. As stated in our emails, we are disassociating presentation type from AC recommendations